# Mamba Can Learn Low-Dimensional Targets In-Context via Test-Time Feature Learning

## Abstract

Mamba, a recently proposed linear-time sequence model, has attracted significant attention for its computational efficiency and strong empirical performance. However, a rigorous theoretical understanding of its underlying mechanisms remains limited. In this work, we provide a theoretical analysis of Mamba's in-context learning (ICL) capability by focusing on tasks defined by low-dimensional nonlinear target functions. Specifically, we study in-context learning of a single-index model $y \approx g_*(\langle \boldsymbol{\beta}, \boldsymbol{x} \rangle)$, which depends on only a single relevant direction $\boldsymbol{\beta}$, referred to as *feature*. We prove that Mamba, pretrained by gradient-based methods, can achieve efficient ICL via *test-time feature learning*, extracting the relevant direction directly from context examples. Consequently, we establish a test-time sample complexity that improves upon linear Transformers—analyzed to behave like kernel methods—and is comparable to nonlinear Transformers, which have been shown to surpass the Correlational Statistical Query (CSQ) lower bound and achieve near information-theoretically optimal rate in previous works. Our analysis reveals the crucial role of the *nonlinear gating* mechanism in Mamba for feature extraction, highlighting it as the fundamental driver behind Mamba's ability to achieve both computational efficiency and high performance.

## 1 Introduction

Mamba (Gu & Dao, 2024), a recently proposed state space model, has rapidly gained attention for its remarkable balance of computational efficiency and empirical performance. By replacing the quadratic-time attention mechanism of Transformers (Vaswani et al., 2017) with a selective state-space recurrence with nonlinear gating, Mamba enables scalable modeling of long sequences while maintaining competitive accuracy across a variety of tasks (Dao & Gu, 2024; Waleffe et al., 2024; Wang et al., 2024; Patro & Agneeswaran, 2025). Despite Mamba's remarkable computational efficiency, it remains unknown whether it can exhibit strong adaptability (often referred to as *feature learning*), a property widely recognized as critical to the success of deep learning neural networks (Girshick et al., 2014; Suzuki, 2019; Damian et al., 2022).

A key benchmark for test-time adaptability is in-context learning (ICL) (Brown et al., 2020), which has emerged as a canonical paradigm for understanding the adaptability of large language models and sequence architectures. By conditioning on context examples provided in the input prompt, a model can achieve strong performance on new tasks at test time without explicit parameter updates. While the empirical effectiveness of ICL is well documented, theoretical understanding of when and how different architectures exhibit this behavior remains limited (Xie et al., 2022; Garg et al., 2022; Zhou et al., 2024). In particular, most existing theoretical analyses focus on Transformers (Ahn et al., 2023; Zhang et al., 2024; Mahankali et al., 2024; Huang et al., 2024; Kim & Suzuki, 2024), whose quadratic attention mechanisms make them both powerful and computationally demanding. It remains unclear whether alternative architectures such as Mamba can offer comparable adaptability.

Recent works have investigated Mamba's ICL capabilities, empirically demonstrating that Mamba performs competitively across various ICL benchmarks (Grazzi et al., 2024; Park et al., 2024; Li et al., 2024c). However, our understanding of Mamba's ICL capabilities remains lacking. This is due to its distinct inductive bias from the Transformer. The recurrent state-space model with nonlinear gating processes inputs through recurrent updates that maintain and transform hidden states over time, rather than relying on global attention over the entire context. This distinction motivates new theoretical questions:

*Can Mamba provably achieve strong test-time adaptability like Transformers*
*with its recurrent state-space updates and nonlinear gating?*

## 1.1 SUMMARY OF CONTRIBUTIONS

In this paper, we study the ICL capabilities of Mamba, focusing on a single-index model—a widely adopted theoretical tool for studying adaptability. We summarize our contributions as follows:

- We introduce a theoretical framework for analyzing Mamba's ICL of single-index models, including input embeddings, the Mamba architecture, and a gradient-based pretraining algorithm. Under this framework, we characterize the optimization dynamics and establish the sample complexity in terms of the number of pretraining tasks and the number of context examples at pretraining and test time required to achieve strong performance (Theorem 3.3).

- Our analysis reveals that pretrained Mamba is capable of *test-time feature learning*, enabling it to extract task-relevant features directly from context examples (Proposition 4.1). This result implies that Mamba can surpass the performance of kernel regression baselines and achieve adaptation at test time. Specifically, the gating mechanism enables Mamba to achieve test-time feature learning, thereby overcoming the limitations inherent to purely linear recurrent updates.

- We provide a comparative analysis between Mamba and Transformer architectures, highlighting similarities and differences in their ICL mechanisms. Our results reveal that Mamba can achieve test-time feature learning via a qualitatively different mechanism—recurrent state-space updates with nonlinear gating—thus extending the theoretical landscape of in-context learning beyond attention-based models.

## 1.2 RELATED WORKS

**Theory of In-Context Learning.** Theoretical investigations of ICL have predominantly centered on Transformers. Beyond initial results showing that Transformers trained on regression tasks can reproduce ordinary least squares solutions in-context (Akyürek et al., 2023; Zhang et al., 2024; Mahankali et al., 2024; Han et al., 2025), subsequent analyses reveal their ability to emulate more complex procedures such as multi-step gradient descent (Ahn et al., 2023; Saunshi et al., 2025), functional gradient descent (Cheng et al., 2024), and sparse regression (Bai et al., 2023). Parallel works extend this line of inquiry to classification, where recent studies provide provable insights into how Transformers implement in-context classification (Li et al., 2024a; Bu et al., 2025).

While the theoretical literature on ICL has dominantly focused on Transformers, a growing body of work is extending this theoretical analysis to linear-time sequence models. Recent works (Li et al., 2024b; 2025b) prove that H3-like model (Fu et al., 2023) and gated linear attention (Yang et al., 2024) can implement weighted preconditioned gradient descent based on loss landscape analysis. Bondaschi et al. (2025) study ICL of Mamba on Markov chains and show that it learns a Laplacian smoothing estimator in-context. However, these works do not provide optimization or generalization guarantees. A recent work by Li et al. (2025a) provides such guarantees for in-context learning of Mamba on classification tasks with outliers, as we also do in this work.

**Learning Low-Dimensional Target Function.** Low-dimensional target function classes, such as sparse parities (Barak et al., 2022; Suzuki et al., 2023; Glasgow, 2024), signal-noise models (Allen-Zhu & Li, 2020; Cao et al., 2022), are widely adopted as theoretical benchmarks for studying a neural network's ability to perform feature learning. This work specifically focuses on the single-index model. A line of theoretical work has analyzed the learning of these models and has established key results on sample complexity. The required sample complexity is governed by either the *information exponent* (for algorithms utilizing correlational information (Arous et al., 2021; Bietti et al., 2022; Damian et al., 2023; Mousavi-Hosseini et al., 2023)) or the *generative exponent* (for algorithms that employ suitable label transformations (Damian et al., 2024; Lee et al., 2024; Arnaboldi et al., 2024; Joshi et al., 2024)). We discuss these sample complexity results in more detail in Section 3.1.

Our work is most closely related to Oko et al. (2024); Nishikawa et al. (2025), which lie at the intersection of ICL and the single-index model. Specifically, Oko et al. (2024) show that a pretrained linear Transformer can effectively learn a single-index model in-context. More recent work by Nishikawa et al. (2025) establish an even smaller sample complexity and reveal that the nonlinear Transformer can perform test-time feature learning. A detailed comparison with these works is provided in Section 3.2.

## 2 PROBLEM SETTING

In this section, we provide a formal description of the key components we focus on: the ICL data distribution, the Mamba model, and the gradient-based pretraining algorithm.

**Notation.** We denote the $i$-th coordinate of a vector $\boldsymbol{v}$ as $\boldsymbol{v}[i]$, and the $(i,j)$-th coordinate of a matrix $\boldsymbol{M}$ as $\boldsymbol{M}[i,j]$. The matrix $\mathrm{diag}(\boldsymbol{v})$ represents a diagonal matrix with a vector $\boldsymbol{v}$ on its main diagonal. We use $\odot$ for the element-wise product. For any $k \in \mathbb{N}$, we denote the vectors with all entries equal to one and zero as $\mathbf{1}_k$ and $\mathbf{0}_k$, respectively. We omit the subscript $k$ when the dimension is clear from the context.

### 2.1 DATA DISTRIBUTION FOR IN-CONTEXT LEARNING

In-context learning aims to solve the *task* of predicting the *label* $y$ of a *query* $\boldsymbol{x}$ by leveraging a sequence of input-label pairs $\{(\boldsymbol{x}_i, y_i)\}_{i \in [N]}$, which are referred to as *context examples*. The model then utilizes a prompt, which is a sequence $(\boldsymbol{x}_1, y_1, \ldots, \boldsymbol{x}_N, y_N, \boldsymbol{x})$ consisting of the context examples and the query, as its input. We focus on the case where prompts are constructed from the Gaussian single-index model, which is defined as follows.

**Definition 2.1** (Gaussian Single-Index Model). Given a feature vector $\boldsymbol{\beta} \in \mathbb{R}^d$, we draw input-label pairs $(\boldsymbol{x}, y) \sim \mathcal{D}_{\boldsymbol{\beta}}$ as

$$\boldsymbol{x} \sim \mathcal{N}(\mathbf{0}, \boldsymbol{I}_d), \quad y = g_*(\langle \boldsymbol{\beta}, \boldsymbol{x} \rangle) + \zeta, \quad \zeta \sim \mathrm{Unif}(\{-\tau, \tau\}),$$

where $g_*$ is a polynomial *link function* and $\tau > 0$ represents the noise level. For simplicity, we assume that $\mathbb{E}_{\mathbf{z} \sim \mathcal{N}(0,1)}[g_*(\mathbf{z})] = 0, \mathbb{E}_{\mathbf{z} \sim \mathcal{N}(0,1)}[(g_*(\mathbf{z}))^2] = 1$ and $\tau$ is a small enough constant.

For each task, a prompt is constructed with a random choice of feature vectors.

**Definition 2.2** (ICL Data Distribution). For given a context length $N$, we define a data distribution $\mathcal{D}(N)$ such that $(\boldsymbol{\beta}, \{(\boldsymbol{x}_i, y_i)\}_{i \in [N]}, \boldsymbol{x}, y) \sim \mathcal{D}(N)$ is constructed as follows.

1. We draw the feature vector $\boldsymbol{\beta} \in \mathbb{R}^d$ uniformly from the unit sphere $S_r$ of a low-dimensional *intrinsic feature space* with dimension $r$, defined as:

$$S_r := \left\{ \boldsymbol{\theta} \in \mathbb{R}^d : \|\boldsymbol{\theta}\| = 1, \boldsymbol{\theta}[j] = 0 \text{ for all } j \notin \mathcal{I} \right\},$$

for some unknown feature index set $\mathcal{I}$ with $|\mathcal{I}| = r$.

2. We sample $N$ context examples $\{(\boldsymbol{x}_i, y_i)\}_{i \in [N]}$ and a query-label pair $(\boldsymbol{x}, y)$ from $\mathcal{D}_{\boldsymbol{\beta}}$.

Our task distribution exhibits a low-dimensional structure in two key aspects: (1) the label depends solely on the projection of the input onto the feature vector, and (2) feature vectors are supported on an $r$-dimensional subspace. We note that to achieve low prediction errors, it is crucial to extract both of these structures and estimate the link function $g_*$.

### 2.2 PREDICTION MODEL ARCHITECTURE

Our prediction model for ICL is composed of three parts: input embedding, one-layer Mamba, multi-layer perceptron (MLP).

**Input Embedding.** Given a prompt $(\boldsymbol{x}_1, y_1, \ldots, \boldsymbol{x}_N, y_N, \boldsymbol{x})$ with context length $N$ and label $y$, we construct an input embedding $\boldsymbol{Z} \in \mathbb{R}^{(\tilde{d}+1) \times (N+1)}$ as

$$\boldsymbol{Z} = \begin{bmatrix} \phi(\boldsymbol{x}_1) & \phi(\boldsymbol{x}_2) & \ldots & \phi(\boldsymbol{x}_N) & \phi(\boldsymbol{x}) \\ y_1 & y_2 & \ldots & y_N & 0 \end{bmatrix} = [\boldsymbol{z}_1, \ldots \boldsymbol{z}_N, \boldsymbol{z}_{N+1}] \in \mathbb{R}^{(\tilde{d}+1) \times (N+1)},$$

where $\tilde{d} = \frac{d(d+1)}{2} + 1$ and $\phi : \mathbb{R}^d \to \mathbb{R}^{\tilde{d}}$ is defined as

$$\phi(\boldsymbol{\theta}) = \left[ 1, \boldsymbol{\theta}[1], \ldots, \boldsymbol{\theta}[d], \frac{\boldsymbol{\theta}[1]^2 - 1}{\sqrt{2}}, \ldots, \frac{\boldsymbol{\theta}[d]^2 - 1}{\sqrt{2}}, \boldsymbol{\theta}[1]\boldsymbol{\theta}[2], \ldots, \boldsymbol{\theta}[d-1]\boldsymbol{\theta}[d] \right].$$

An input embedding similar to ours was also considered in the recent work by Sun et al. (2025), who studied the in-context learning of high-order polynomial target functions. They showed this embedding can be implemented with a simple version of Gated Linear Unit (GLU) and demonstrated its efficacy for enabling linear Transformers to learn these functions in-context. Unlike Sun et al. (2025), who repeatedly stacked a linear Transformer and a GLU layer, in our work, a single GLU-based embedding is sufficient due to the nonlinearity in Mamba and MLP layers. We discuss the efficacy of this input embedding in more detail in Section 4.1.

*Remark* 2.3. Our input embedding is based on a basis for degree-2 polynomials in $\mathbb{R}^d$. Specifically, we use the standard basis of $\mathbb{R}^d$ for the construction of both the input embedding and the intrinsic feature space $S_r$. While extending our results to a general choice of $S_r$ with an arbitrary basis may require additional techniques, our setting remains valuable for studying Mamba's ability to learn low-dimensional structure. Furthermore, we emphasize that our result also holds with the standard choice of input embedding $\phi(\boldsymbol{x}) = \boldsymbol{x}$ with $\tilde{d} = d$, as considered in prior works including Von Oswald et al. (2023), for the case where link function $g_*$ is not an even function. We refer to Section 4 for a more detailed discussion.

**One-Layer Mamba.** Given an input embedding $\boldsymbol{Z} = (\boldsymbol{z}_1, \ldots, \boldsymbol{z}_{N+1}) \in \mathbb{R}^{(\tilde{d}+1) \times (N+1)}$, a one-layer Mamba model $\mathsf{Mamba}(\cdot; \boldsymbol{\Theta})$ with parameters $\boldsymbol{\Theta}$ has sequential outputs $\boldsymbol{o}_1, \ldots, \boldsymbol{o}_{N+1} \in \mathbb{R}^{\tilde{d}+1}$ and hidden states for $i$-th channel $\boldsymbol{h}_1^{(i)}, \ldots, \boldsymbol{h}_N^{(i)} \in \mathbb{R}^{d_h}$ defined as below:

$$\boldsymbol{h}_l^{(i)} = \overline{\boldsymbol{A}}_l \boldsymbol{h}_{l-1}^{(i)} + \overline{\boldsymbol{B}}_l \boldsymbol{z}_l[i] \in \mathbb{R}^{d_h}, \quad \boldsymbol{o}_l[i] = \boldsymbol{C}_l^\top \boldsymbol{h}_l^{(i)} \in \mathbb{R},$$

$$\overline{\boldsymbol{A}}_l = \exp\left(\Delta_l \boldsymbol{A}\right) \in \mathbb{R}^{d_h \times d_h}, \quad \overline{\boldsymbol{B}}_l = (\Delta_l \boldsymbol{A})^{-1}\left(\exp\left(\Delta_l \boldsymbol{A}\right) - \boldsymbol{I}_{d_h}\right)\Delta_l \boldsymbol{B}_l \in \mathbb{R}^{d_h},$$

where $\boldsymbol{h}_0^{(i)} = \boldsymbol{0}_{d_h}$ and $\boldsymbol{A} \in \mathbb{R}^{d_h \times d_h}$. Here, the components of the selection algorithm $\boldsymbol{A}, \boldsymbol{B}_l, \boldsymbol{C}_l, \Delta_l$ is chosen as

$$\boldsymbol{A} = -\boldsymbol{I}_{\tilde{d}+1}, \quad \boldsymbol{B}_l = \boldsymbol{W}_B \boldsymbol{z}_l, \quad \boldsymbol{C}_l = \boldsymbol{W}_C \boldsymbol{z}_l, \quad \Delta_l = \mathrm{softplus}\left(\boldsymbol{w}^\top \boldsymbol{z}_l + b\right),$$

with parameters $\boldsymbol{W}_B, \boldsymbol{W}_C \in \mathbb{R}^{d_h \times (\tilde{d}+1)}, \boldsymbol{w} \in \mathbb{R}^{\tilde{d}+1}, b \in \mathbb{R}$. Then, the $l$-th output can be expressed as

$$\boldsymbol{o}_l = \sum_{j=1}^{l} G_{j,l}(\boldsymbol{Z}) \boldsymbol{z}_j \boldsymbol{z}_j^\top \boldsymbol{W}_B^\top \boldsymbol{W}_C \boldsymbol{z}_l, \tag{1}$$

where $G_{j,l}(\boldsymbol{Z}) = \sigma\left(\boldsymbol{w}^\top \boldsymbol{z}_j + b\right) \prod_{k=j+1}^{l}\left(1 - \sigma\left(\boldsymbol{w}^\top \boldsymbol{z}_k + b\right)\right)$ with sigmoid function $\sigma(\cdot)$. It implies that Mamba involves two key mechanisms: *nonlinear gating* $G_{j,l}(\boldsymbol{Z})$ and *linear attention* with projection matrices $\boldsymbol{W}_B$ and $\boldsymbol{W}_C$. Yang et al. (2024) refer the combination of these mechanisms as *gated linear attention* and recent recurrent models including Mamba, mLSTM (Beck et al., 2024), and RWKV-6 (Peng et al., 2024) can be viewed within this framework.

To ensure a tractable optimization guarantee, we further introduce the following simplifications to our model:

$$\boldsymbol{W}_B^\top \boldsymbol{W}_C = \mathrm{diag}(\boldsymbol{\gamma}, 0), \quad \boldsymbol{w} = \begin{bmatrix} \boldsymbol{0}_{\tilde{d}} \\ \rho^{-1} \end{bmatrix},$$

where $\boldsymbol{\gamma} \in \mathbb{R}^{\tilde{d}}$ is a learnable parameter, while $\boldsymbol{w} \in \mathbb{R}^{\tilde{d}+1}$ and $b \in \mathbb{R}$ are fixed. Our approach of merging the product of two learnable matrices into a single matrix and using sparse learnable parameters is a technique also adopted in the theoretical literature on optimization of attention mechanisms (Ahn et al., 2023; Zhang et al., 2024; Mahankali et al., 2024; Kim & Suzuki, 2024). Under this simplification, the last coordinate of the final output which serves as the input to the MLP can be expressed as follows:

$$\mathsf{Mamba}(\boldsymbol{Z}; \boldsymbol{\gamma})[\tilde{d}+1, N+1] = \sum_{j=1}^{N} G_{j,N+1}(\boldsymbol{Z}) y_j \phi(\boldsymbol{x}_j)^\top \left(\boldsymbol{\gamma} \odot \phi(\boldsymbol{x})\right).$$

**Multi-Layer Perceptron.** We use a two-layer MLP with ReLU activation, width $m$ and parameters $\boldsymbol{u}, \boldsymbol{v}, \boldsymbol{a} \in \mathbb{R}^m$ defined as follows:

$$\mathsf{MLP}(z; \boldsymbol{u}, \boldsymbol{v}, \boldsymbol{a}) := \sum_{k=1}^{m} \boldsymbol{u}[k] \mathrm{ReLU}\left(\boldsymbol{v}[k]z + \boldsymbol{a}[k]\right).$$

We apply this MLP to the output of the Mamba layer, after normalizing it by its context length $N$. Then, the final output is given by

$$f(\boldsymbol{Z}; \boldsymbol{\gamma}, \boldsymbol{u}, \boldsymbol{v}, \boldsymbol{a}) := \mathsf{MLP}\left(N^{-1}\mathsf{Mamba}(\boldsymbol{Z}; \boldsymbol{\gamma})[\tilde{d}+1, N+1]; \boldsymbol{u}, \boldsymbol{v}, \boldsymbol{a}\right)$$

$$= \sum_{k=1}^{m} \boldsymbol{u}[k] \mathrm{ReLU}\left(\boldsymbol{v}[k] N^{-1} \sum_{j=1}^{N} G_{j,N+1}(\boldsymbol{Z}) y_j \phi(\boldsymbol{x}_j)^\top \left(\boldsymbol{\gamma} \odot \phi(\boldsymbol{x})\right) + \boldsymbol{a}[k]\right).$$

*Remark* 2.4. A similar structure to our models, which combines a sequence model with a MLP, has also been utilized in two closely related prior works. For example, Nishikawa et al. (2025) follow a similar structure but use a softmax Transformer in place of Mamba. In contrast, Oko et al. (2024) use a different architectural design, applying the MLP to the input embedding before a linear Transformer, rather than after the sequence model.

Our goal for ICL is to find parameters $\boldsymbol{\gamma} \in \mathbb{R}^{\tilde{d}}$, $\boldsymbol{u}, \boldsymbol{v}, \boldsymbol{a} \in \mathbb{R}^m$, and context length $N$, achieving a small ICL test error, which is defined as

$$\mathcal{R}_N(\boldsymbol{\gamma}, \boldsymbol{u}, \boldsymbol{v}, \boldsymbol{a}) := \mathbb{E}_{(\boldsymbol{Z}, y) \sim \mathcal{D}(N)}[|f(\boldsymbol{Z}, \boldsymbol{\gamma}, \boldsymbol{u}, \boldsymbol{v}, \boldsymbol{a}) - y|].$$

Here, we abuse notation and use $(\boldsymbol{Z}, y)$ to denote the input embedding and label for a prompt sampled from the ICL data distribution $\mathcal{D}(N)$. More precisely, we are interested in the sample complexity of context examples required for the parameters learned from pretraining to achieve a low ICL test error.

## 2.3 Pretraining Algorithm

Our prediction model is pretrained on a set of $T_{\mathrm{pt}} = T_1 + T_2$ tasks with context length $N_{\mathrm{pt}}$ drawn from $\mathcal{D}(N_{\mathrm{pt}})$. For each task $t \in [T_{\mathrm{pt}}]$, let we have input embedding $\boldsymbol{Z}^t$ constructed from context examples $\{(\boldsymbol{x}_i^t, y_i^t)\}_{i \in [N_{\mathrm{pt}}]}$ and a query-label pair $(\boldsymbol{x}^t, y^t)$ with a feature vector $\boldsymbol{\beta}^t$. Then, our training losses can be written as

$$L_l(\boldsymbol{\gamma}, \boldsymbol{u}, \boldsymbol{v}, \boldsymbol{a}) := \frac{1}{T_l} \sum_{t=T_{l-1}+1}^{T_{l-1}+T_l} \left( f\left(\boldsymbol{Z}^t; \boldsymbol{\gamma}, \boldsymbol{u}, \boldsymbol{v}, \boldsymbol{a}\right) - y^t \right)^2,$$

for $l = 1, 2$ and $T_0 = 0$. We employ a two-stage training procedure, as described in Algorithm 1, using these objectives.

1. In Stage I, we only train the Mamba layer parameter $\boldsymbol{\gamma}$, starting from proper initialization. Our training objective is $\ell_2$-regularized loss $L_1(\boldsymbol{\gamma}, \boldsymbol{u}, \boldsymbol{v}, \boldsymbol{a}) + \frac{\lambda_1}{2} \|\boldsymbol{\gamma}\|^2$. Due to the non-linearity introduced by the MLP, this objective is non-convex. To make the training dynamics tractable, we apply one-step gradient descent, following the approaches studied in the literature of feature learning (Ba et al., 2022; Damian et al., 2022). As we describe in Section 4.1, a single step update is sufficient to capture the low-dimensional structure of the feature vectors.

2. In Stage II, we fix the Mamba layer parameter $\boldsymbol{\gamma}^*$ obtained from Stage I and optimize the outer layer $\boldsymbol{u}$ of MLP on $\ell_2$-regularized loss $L_2(\boldsymbol{\gamma}^*, \boldsymbol{u}, \boldsymbol{v}^*, \boldsymbol{a}^*) + \frac{\lambda_2}{2} \|\boldsymbol{u}\|^2$ with reinitialized inner layer parameters $\boldsymbol{v}^*, \boldsymbol{a}^*$. This induces a convex problem that gradient-based methods can solve. As we show in Section 4.2, the optimized MLP is capable of estimating the link function $g_*$.

---

**Algorithm 1:** Gradient-based Pretraining of the Mamba Model

---

**Input** : Learning rate $\eta$, weight decay $\lambda_1, \lambda_2$, context length $N_{\mathrm{pt}}$, the number of tasks $T_1, T_2$, initialization scale $\gamma, \rho, b$.

**Stage I: Gradient descent on Mamba layer**

  Initialize $\boldsymbol{\gamma} = (\gamma^2, 1, \ldots, 1, \gamma, \cdots, \gamma)$, $\boldsymbol{u}(0) = m^{-1}\boldsymbol{1}_m$, $\boldsymbol{v}(0) = \boldsymbol{1}_m$, $\boldsymbol{a}(0) = \boldsymbol{0}_m$.

  $\boldsymbol{\gamma}^* \leftarrow \boldsymbol{\gamma}^{(0)} - \eta \nabla_{\boldsymbol{\gamma}} \left( L_1(\boldsymbol{\gamma}(0), \boldsymbol{u}(0), \boldsymbol{v}(0), \boldsymbol{a}(0)) + \frac{\lambda_1}{2} \|\boldsymbol{\gamma}\|^2 \right).$

**Stage II: Optimization of MLP Layer**

  Initialize $\boldsymbol{v}^* \sim \mathrm{Unif}(\{\pm 1\}^m)$, $\boldsymbol{a}^* \sim \mathrm{Unif}([-1, 1]^m)$.

  $\boldsymbol{u}^* \leftarrow \arg\min_{\boldsymbol{u}} \left( L_2(\boldsymbol{\gamma}^*, \boldsymbol{u}, \boldsymbol{v}^*, \boldsymbol{a}^*) + \frac{\lambda_2}{2} \|\boldsymbol{u}\|^2 \right).$

**Output** : Prediction function $f(\cdot; \boldsymbol{\gamma}^*, \boldsymbol{u}^*, \boldsymbol{v}^*, \boldsymbol{a}^*)$.

---

## 3 Mamba Efficiently Learns Single-Index Models In-Context

In this section, we present our theoretical results on the ICL performance of our model. Our analysis focuses on the asymptotic dependencies on the input dimension $d$, with the assumption that the feature dimension $r$ can scale with $d$, while the link function $g_*$ is fixed. For our analysis, we let $N^*$ and $T^*$ be the maximum admissible context length and the number of pretraining tasks, respectively.

We assume that $N^*, T^* \le d^{C^*}$ for some large constant $C^* > 0$. We use the standard asymptotic notation $\mathcal{O}(\cdot), \Omega(\cdot), \Theta(\cdot), o(\cdot)$ to express dependencies on $d$, and $\tilde{\mathcal{O}}(\cdot), \tilde{\Omega}(\cdot), \tilde{\Theta}(\cdot)$ to hide logarithmic factors of $d$.

## 3.1 PRELIMINARIES

We first provide backgrounds on learning Gaussian single-index models, which are essential for understanding our main result. Let $\text{He}_i(z) = (-1)^i e^{\frac{z^2}{2}} \frac{d^i}{dz^i} e^{-\frac{z^2}{2}}$ denote the probabilist's Hermite polynomials. Then, the set $\{\text{He}_i(z)/\sqrt{i!}\}_{i \in \mathbb{N} \cup \{0\}}$ forms an orthonormal basis of the $L^2$ space with respect to the Gaussian measure and serves as a key technical tool for the analysis of Gaussian single-index models. We now introduce two key terms relevant to the sample complexity of learning.

**Definition 3.1.** For any function $h : \mathbb{R} \to \mathbb{R}$ which is $L^2$-integrable with respect to the Gaussian measure, we express its Hermite expansion as

$$h(z) = \sum_{i=0}^{\infty} \frac{H(h, i)}{i!} \text{He}_i(z), \quad H(h, i) := \mathbb{E}_{\mathbf{z} \sim \mathcal{N}(0,1)}[h(\mathbf{z})\text{He}_i(\mathbf{z})].$$

We also define the following quantities:

- We define $\deg(h)$ as the degree of $h$, if it is a polynomial.

- The *information exponent* (Arous et al., 2021; Damian et al., 2023) of $h$ is defined as

$$\text{ie}(h) := \min\{i \in \mathbb{N} : H(h, i) \ne 0\}.$$

  It implies that $\mathbb{E}_{\mathbf{z} \sim \mathcal{N}(0,1)}[h(\mathbf{z})\text{He}_k(\mathbf{z})] = 0$ for any $k \in \mathbb{N}$ with $k < \text{ie}(h)$.

- The *generative exponent* (Damian et al., 2024) of $h$ is defined as the lowest possible information exponent after an $L^2$ transformation. It is formally defined as:

$$\text{ge}(h) := \min_{\mathcal{T} \in L^2(\mathbb{P}_h)} \min\{i \in \mathbb{N} : H(\mathcal{T} \circ h, i) \ne 0\},$$

  where $L^2(\mathbb{P}_h)$ is the set of $L^2$-integrable functions with respect to $\mathbb{P}_h$. Here, $\mathbb{P}_h$ is the push-forward measure of the Gaussian measure by $h$.

While the definition of the generative exponent may seem difficult to apply at first glance, Lee et al. (2024) provides a characterization of the generative exponent for polynomials.

**Lemma 3.2** (Proposition 6 in Lee et al. (2024)). *For a polynomial link function $g_*$, the generative exponent is characterized as $\text{ge}(g_*) = 2$ if $g_*$ is an even function, and $\text{ge}(g_*) = 1$ otherwise.*

From the definition, $\text{ge}(g_*) \le \text{ie}(g_*) \le \deg(g_*)$ and Lemma 3.2 implies that the gap between these three terms can be arbitrarily large depending on the choice of $g_*$[1]. With a slight abuse of notation, we use $\Theta(\deg(g_*))$ to denote a quantity that is bounded by a universal constant multiple of $\deg(g_*)$. We also use $\Theta(\text{ie}(g_*))$ and $\Theta(\text{ge}(g_*))$, in similar manners.

**Sample Complexity of Learning Single-Index Models.** Previous works have established the sample complexity of various methods for learning a Gaussian single-index model. For example, kernel methods, which lack an adaptive basis, require at least $d^{\deg(g_*)}$ samples (Ghorbani et al., 2021; Donhauser et al., 2021). In contrast, adaptive methods such as gradient-based methods on two-layer neural networks can achieve a sample complexity of $\tilde{\mathcal{O}}\left(d^{\Theta(\text{ie}(g_*))}\right)$ by learning an adaptive feature map (Arous et al., 2021; Ba et al., 2022; Damian et al., 2022; 2023; Dandi et al., 2024). These approaches fall under the category of CSQ algorithms, and in this category, a sample complexity that depends on the information exponent is inevitable (Damian et al., 2022). However, recent works show that a nonlinear transformation introduced by data reuse (Arnaboldi et al., 2024; Lee et al., 2024) enables the algorithm to move into the broader class of Statistical Query (SQ) algorithms. This transformation allows the "effective" information exponent to be lowered to the generative exponent, thereby achieving a sample complexity of $\tilde{\mathcal{O}}\left(d^{\Theta(\text{ge}(g_*))}\right)$.

---

[1]For example, consider $g_*(z) = \text{He}_q(z) + \text{He}_p(z)$ with $1 \ll q \ll p$.

## 3.2 MAIN RESULT

We now present our main result, which provides a theoretical characterization of the pretraining and test-time sample complexities for achieving low ICL errors.

**Theorem 3.3.** *Let $f(\cdot; \boldsymbol{\gamma}^*, \boldsymbol{u}^*, \boldsymbol{v}^*, \boldsymbol{a}^*)$ be the Mamba model pretrained using Algorithm 1. We assume the following conditions hold for its hyperparameters:*

- *The context length is $N_{\text{pt}} = \tilde{\Omega}\left(\max\left\{r^{3\text{ge}(g_*)}d^8, T_1^2 d^4\right\}\right)$.*

- *The number of pretraining tasks are $T_1 = \tilde{\Omega}\left(r^{3\text{ge}(g_*)}d^6\right)$ and $T_2 = \tilde{\Omega}\left(r^{3\text{ge}(g_*)}\right)$.*

- *The MLP width is $m = \tilde{\Omega}\left(r^{4\text{ge}(g_*)}\right)$.*

- *The fixed weights are $\rho = \Theta\left((\log d)^{C_\rho}\right)$ and $b = C_b \log d$, and the initialization scale is $\gamma = \Theta((\log d)^{-C_\gamma})$ for sufficiently large constants $C_\gamma, C_\rho, C_b > 0$.*

*Then, there exist hyperparameters $\lambda_1, \lambda_2,$ and $\eta$ such that with probability at least 0.99 over the training data and random initialization, the following holds: If the test prompt length satisfies $N_{\text{test}} = \tilde{\Omega}\left(r^{3\text{ge}(g_*)}\right)$, then the test error $\mathcal{R}_{N_{\text{test}}}(\boldsymbol{\gamma}^*, \boldsymbol{u}^*, \boldsymbol{v}^*, \boldsymbol{a}^*)$ is bounded by $\tau + o(1)$.*

We discuss our sample complexity results in comparison with other methods, including regression on test prompts and prior theoretical works (Oko et al., 2024; Nishikawa et al., 2025). We summarize these results in Table 1 and highlight the following key points:

**Adaptation to Low-Dimensional Structure.** Our sample complexity depends on the intrinsic dimension of the feature vectors $r$, rather than the ambient dimension $d$. This is consistent with prior works (Oko et al., 2024; Nishikawa et al., 2025) that also demonstrate a dependence on intrinsic dimensionality. In contrast, the sample complexities of various regression algorithms we have discussed depend on the full input dimension $d$. This difference arises because pretrained models can learn the low-dimensional structure of the intrinsic feature space during pretraining.

**Test-Time Feature Learning.** The dependence of the sample complexity on the intrinsic dimension $r$ in the work of Oko et al. (2024) is controlled by the degree of the link function $g_*$. This means that while their approach is more efficient than simple regression on full dimensions, its performance remains close to that of kernel methods on intrinsic dimensions. In contrast, our result depends on the generative exponent $\text{ge}(g_*)$, instead of its degree. This implies that Mamba's efficient in-context learning is enabled not just by its ability to learn an intrinsic feature space, but also by a process called *test-time feature learning*, which allows the model to extract features directly from the context. The same process also works for the softmax Transformers considered in Nishikawa et al. (2025) and thus achieved a similar sample complexity. However, these models perform test-time feature learning through different mechanisms: Mamba relies on nonlinear gating, while the Transformer uses softmax attention.

**Improvement in Pretraining Sample Complexity.** The conditions for the pretraining in our theorem can be satisfied with a pretraining sample complexity of $N_{\text{pt}} = \Theta\left(d^{\Theta(\text{ge}(g_*))}\right)$. In contrast, the pretraining sample complexities in previous works (Oko et al., 2024; Nishikawa et al., 2025) are governed by the information exponent, which can lead to a suboptimal rate in the worst case. This improvement is due to the nonlinearity of the MLP, as we discuss in detail in Section 4.

| Regression on Test Prompt | | |
|:---:|:---:|:---:|
| Kernel | CSQ | SQ |
| $d^{\Theta(\deg(g_*))}$ | $d^{\Theta(\text{ie}(g_*))}$ | $d^{\Theta(\text{ge}(g_*))}$ |
| **In-context learning** | | |
| Linear Transformer | Softmax Transformer | Mamba |
| Oko et al. (2024) | Nishikawa et al. (2025) | This Work |
| *Pretrain:* $d^{\Theta(\text{ie}(g_*))}$ *Test:* $r^{\Theta(\deg(g_*))}$ | *Pretrain:* $d^{\Theta(\text{ie}(g_*))}$ *Test:* $r^{\Theta(\text{ge}(g_*))}$ | *Pretrain:* $d^{\Theta(\text{ge}(g_*))}$ *Test:* $r^{\Theta(\text{ge}(g_*))}$ |

**Table 1:** Summary of sample complexity results for regression algorithms on test prompt and prior works on in-context learning (Oko et al., 2024; Nishikawa et al., 2025).

# 4 PROOF OVERVIEW

In this section, we provide an overview of the proof for our theorem. The proof consists of three main parts: an analysis of one-step gradient descent on the Mamba layer, the optimization of the MLP, and a test error analysis. The formal proofs for each of these steps are provided in Appendices B, C, and D, respectively. In the following, we introduce the key ideas behind each step.

## 4.1 ONE-STEP GRADIENT DESCENT ON THE MAMBA LAYER

We show that the pretrained parameter $\boldsymbol{\gamma}^*$ recovers the intrinsic feature space $S_r$ by attaining significantly larger components within the feature index set $\mathcal{I}$ than in other indices. Furthermore, we show that pretrained Mamba performs test-time feature learning by establishing the following proposition (formally stated in Proposition B.5):

**Proposition 4.1** (Informal). *For a sampled ICL input embedding $\boldsymbol{Z}$ with context length $N = \tilde{\Omega}\left(r^{3\mathrm{ge}(g_*)}\right)$, query $\boldsymbol{x}$, and feature vector $\boldsymbol{\beta}$, the following holds with high probability:*

$$\mathsf{Mamba}(\boldsymbol{Z}; \boldsymbol{\gamma}^*) \approx P_1 + P_2 \left(\frac{\langle \boldsymbol{\beta}, \boldsymbol{x} \rangle}{r}\right)^{\mathrm{ge}(g_*)}, \tag{2}$$

*where $P_1$ and $P_2$ are independent of the data.*

Assuming a negative bias $b$ with sufficiently large absolute value, and a large enough number of tasks $T_1$ and context length $N_{\mathrm{pt}}$, the updated parameter $\boldsymbol{\gamma}^*$ can be approximated as follows:

$$\boldsymbol{\gamma}^* \approx 2\eta \mathbb{E}_{(\boldsymbol{Z},y)\sim\mathcal{D}(N_{\mathrm{pt}})}[y\nabla_{\boldsymbol{\gamma}} f(\boldsymbol{Z}; \boldsymbol{\gamma}(0), \boldsymbol{u}(0), \boldsymbol{v}(0), \boldsymbol{a}(0))]$$
$$\approx 2\eta \mathbb{E}_{\substack{\boldsymbol{\beta}\sim\mathrm{Unif}(S_r) \\ (\boldsymbol{x},y)\sim\mathcal{D}_{\boldsymbol{\beta}}}} \left[y\mathbb{1}\left[\langle \boldsymbol{c}_{\boldsymbol{\beta}}, \boldsymbol{\gamma}(0) \odot \phi(\boldsymbol{x}) \rangle > 0\right] \boldsymbol{c}_{\boldsymbol{\beta}} \odot \phi(\boldsymbol{x})\right],$$

where $\boldsymbol{c}_{\boldsymbol{\beta}} := \mathbb{E}_{(\boldsymbol{x},y)\sim\mathcal{D}_{\boldsymbol{\beta}}}[y\sigma(y/\rho + b)\phi(\boldsymbol{x})]$ corresponds to a simplified expectation over context examples, neglecting the effect of "forgetting" in the gating mechanism.

**The Role of Gating and Input Embedding.** For the proof of Proposition 4.1 and our test-time sample complexity, nonlinear transformation introduced by gating mechanism plays a crucial role. In the absence of a gating mechanism and with only a linear attention, the term $\boldsymbol{c}_{\boldsymbol{\beta}}$ is replaced by $\mathbb{E}_{(\boldsymbol{x},y)\sim\mathcal{D}_{\boldsymbol{\beta}}}[y\phi(\boldsymbol{x})]$ and this term vanishes when $\mathrm{ie}(g_*) > 2$. This is a consequence of our input embedding using Hermite polynomials only up to the second degree, in combination with Stein's lemma. As a result, pretraining in this case is unable to learn useful information. However, we prove that the gating mechanism ensures $c_{\boldsymbol{\beta}}$ is non-zero, thereby enabling the model to extract information. This is because the nonlinear transformation introduced by the gating mechanism reduces the information exponent to the generative exponent: $\mathrm{ie}(g_*\sigma(g_*/\rho + b)) = \mathrm{ge}(g_*)$. This reduction, combined with Lemma 3.2 and our input embedding, makes information extraction possible. When $g_*$ is a non-even function, our result can also be shown to hold with the standard input embedding $\phi(\boldsymbol{x}) = \boldsymbol{x}$, as can be seen from this intuition. Reducing the information exponent to the generative exponent crucially affects the achievement of a test-time sample complexity below the CSQ lower bounds. Nishikawa et al. (2025) shows that the softmax operator in the Transformer can also perform a nonlinear transformation on the label, which reduces the information exponent. This highlights a key difference in the mechanisms used by Mamba and the softmax Transformer to achieve this result.

**Improved Sample Complexity of Pretraining.** The nonlinearity of the MLP is crucial for our analysis of the pretraining sample complexity. If the indicator $\mathbb{1}[\cdot]$ inside the expectation is 1 with high probability, then the updated parameter $\boldsymbol{\gamma}^*$ can be approximated as $\boldsymbol{\gamma}^* \approx 2\eta\mathbb{E}_{\boldsymbol{\beta}\sim\mathrm{Unif}(S_r)}\left[\mathbb{E}_{(\boldsymbol{x},y\sim\mathcal{D}_{\boldsymbol{\beta}})}[y\boldsymbol{c}_{\boldsymbol{\beta}} \odot \phi(\boldsymbol{x})]\right]$ and this close to zero when $\mathrm{ie}(g_*) > 2$, leading to less information gain. However, we show that the indicator deviates significantly from a constant value. We formally prove that this deviation allows the indicator to reduce the information exponent when multiplied by the label $y$, thereby inducing a pretraining sample complexity not governed by $\mathrm{ie}(g_*)$. While Nishikawa et al. (2025) employ an architecture with a similar structure to ours—an MLP following a Softmax Transformer—they do not achieve the same improvement. This is because their use of Softmax places the model in a regime where a key indicator function is 1 with high probability, which is sufficient in their case as Softmax generates the necessary higher-order functions of the input, while our analysis only uses up to second-order functions. In addition, our observation for this improvement cannot be directly applied to the work of Oko et al. (2024) due to a key architectural difference: applying the MLP layer in the input embedding rather than at the output layer.

### 4.2 OPTIMIZATION OF THE MLP AND TEST ERROR ANALYSIS

In our analysis of Stage II pretraining, we show that the MLP can fit the link function $g_*$. We first construct an outer layer parameter $\boldsymbol{u}'$ such that the loss $L_2(\boldsymbol{\gamma}^*, \boldsymbol{u}', \boldsymbol{v}, \boldsymbol{a})$ is sufficiently small and the norm $\|\boldsymbol{u}'\|$ is well-bounded. Our construction is based on the techniques in Damian et al. (2022), which constructed a ReLU network approximating monomials. This allows our model to learn high-order polynomials with a few layers, in contrast to the multi-layer approach in Sun et al. (2025). This is possible because Lemma 3.2 implies that $g_*(z)$ is a polynomial of $z^{\mathrm{ge}(g_*)}$, allowing us to construct an MLP that approximates $g_*(\langle \boldsymbol{\beta}, \boldsymbol{x} \rangle)$, when the input is provided in the form of (2).

From the equivalence between $\ell_2$-regularization and $\ell_2$ norm-constraints in convex problems, we show that for a proper $\lambda_2 > 0$, the minimizer $\boldsymbol{u}^*$ satisfies $L_2(\boldsymbol{\gamma}^*, \boldsymbol{u}^*, \boldsymbol{v}^*, \boldsymbol{a}^*) \le L_2(\boldsymbol{\gamma}^*, \boldsymbol{u}', \boldsymbol{v}^*, \boldsymbol{a}^*)$ and $\|\boldsymbol{u}^*\| \le \|\boldsymbol{u}'\|$. Next, we show that the trained model achieves a small test error with context length $N_{\mathrm{pt}}$ by applying a standard generalization bound based on Rademacher complexity, which is applicable due to a well-bounded norm $\|\boldsymbol{u}^*\|$. Lastly, we extend this error bound to a general context length $N_{\mathrm{test}} = \tilde{\Omega}\left(r^{3\mathrm{ge}(g_*)}\right)$. It is possible because (2) implies that prompts with $N_{\mathrm{test}}$ context examples and $N_{\mathrm{pt}}$ context examples give similar outputs, given the same query.

## 5 EXPERIMENTS

To support our theoretical findings, we pretrain and evaluate both Transformer and Mamba models on our data distribution. Our base configuration uses a link function $g_*(z) = \mathrm{He}_3(z)/\sqrt{6}$, an intrinsic dimension of $r = 8$, and an ambient dimension $d = 32$. We employ a 6-layer GPT-2 model (Radford et al., 2019) with 8 attention heads and a 12-layer Mamba model. To ensure a fair comparison, both models have an embedding dimension of 256 and a similar number of parameters. The overall experimental settings for pretraining follow those of prior works (Garg et al., 2022; Park et al., 2024) including a pretraining context length $N_{\mathrm{pt}} = 64$. We also conduct kernel ridge regression on the intrinsic feature space to serve as a baseline for understanding the effect of feature learning. For this, we use a Gaussian RBF kernel with a bandwidth of 1 and a ridge parameter of 1. For evaluation, we measure the prediction error using squared error, with the number of context examples ranging from 1 to 40. We estimate the test error on 1024 randomly sampled tasks, using 2048 independent prompts for each task, and represent the results with the mean and standard deviation over these tasks.

To validate our theoretical results on how problem parameters influence performance, we then analyze trends by varying parameters from our base configuration: the ambient dimension to $d = 16$, the intrinsic dimension to $r = 16$, and the pretraining context length to $N_{\mathrm{pt}} = 16$. Figure 1a demonstrates the influence of the ambient dimension $d$. Both Transformer and Mamba models exhibit comparable performance that is rarely affected by the ambient dimension $d$. In contrast, when the intrinsic dimension $r$ is increased, both models exhibit performance degradation, while their performance remains comparable (Figure 1b). This suggests that both models mainly utilize information from the intrinsic feature space. In addition, these methods outperform kernel methods, even when we restrict the input of the kernel method to the intrinsic feature space. This observation aligns with our finding that Mamba, similar to Transformers, not only benefits from its adaptation to the intrinsic feature space but also performs test-time feature learning. Lastly, we observe different behavior when using a small pretraining context length $N_{\mathrm{pt}} = 16$. In this case, the Transformer's performance deteriorates significantly, while Mamba's performance remains comparable to its performance with $N_{\mathrm{pt}} = 64$. This observation aligns with our pretraining sample complexity result, which is lower than that of the Transformer established by Nishikawa et al. (2025).

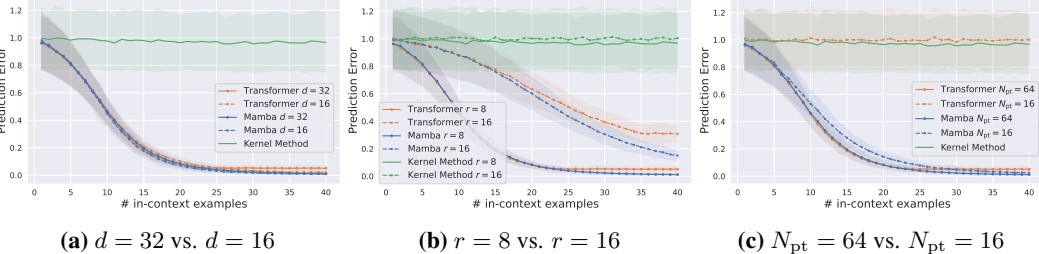

**(a)** $d = 32$ vs. $d = 16$    **(b)** $r = 8$ vs. $r = 16$    **(c)** $N_{\mathrm{pt}} = 64$ vs. $N_{\mathrm{pt}} = 16$

**Figure 1:** Comparison of prediction error for in-context learning with Transformer and Mamba models, and kernel regression across different problem parameter settings.

## 6 CONCLUSION

We investigated Mamba's capability for in-context learning by focusing on a Gaussian single-index model. We proved that Mamba, when pretrained with gradient-based optimization, can efficiently learn in-context through a mechanism we termed test-time feature learning. Our derived test-time sample complexity is comparable to that of the softmax Transformer model, a result established by Nishikawa et al. (2025) and also surpasses the CSQ lower bound. Our analysis reveals that Mamba's gating mechanism is a key factor in enabling feature learning and strong performance. We also presented experimental results to support our findings.

We suggest several directions for future research. First, a valuable direction is to investigate whether our results can be extended to more general input embeddings by considering additional layers, which could help overcome our current limitations. Second, while our analysis considers the case where "forgetting" in the gating mechanism is negligible, recent work by Li et al. (2025a) reveals that this effect can be beneficial for tasks with outliers. Investigating the combination of this effect with our insight could be an interesting direction. Finally, studying how different choices of gating functions within the gated linear attention framework (Yang et al., 2024) lead to different behaviors is a possible direction for future work.

## DECLARATION OF LLM USAGE

LLMs were used solely for editing and refining the writing, including correcting grammar and improving sentence structure. They were not used to generate any original content or ideas.

## ETHICS AND REPRODUCIBILITY STATEMENTS

This work is mainly theoretical and has no ethical concerns. For reproducibility, we state all assumptions and limitations and include full proofs in the appendix.

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

CONTENTS

## A PROOF PRELIMINARIES

**Notation.** We introduce the following additional notation for ease of presentation. We use $\mathbb{1}[\cdot]$ to represent indicator function. We write $a \lesssim b$, $a \gtrsim b$, and $a \asymp b$ to denote that $a = \mathcal{O}(b)$, $a = \Omega(b)$, and $a = \Theta(b)$, respectively. We also use $\mathrm{poly}(d)$ and $\mathrm{polylog}(d)$ to represent a sufficiently large polynomial in $d$ and $\log d$, respectively. Lastly, we use $o(1/\mathrm{polylog}(d))$ to represent a quantity that decreases faster than $(\log d)^{-C}$ for any constant $C > 0$. Lastly, with a slight abuse of notation, we use the asymptotic notation we introduced to represent a vector when its norm satisfies the corresponding bound.

### A.1 SIMPLIFICATION OF MAMBA OUTPUT

The following lemma immediately implies (1).

**Lemma A.1.** *Given a prompt* $(\boldsymbol{x}_1, y_1, \ldots, \boldsymbol{x}_N, y_N, \boldsymbol{x})$ *and its input embedding* $\boldsymbol{Z} \in \mathbb{R}^{(\tilde{d}+1)\times(N+1)}$. *Let* $\mathsf{Mamba}(\boldsymbol{Z}; \boldsymbol{\Theta}) = (\boldsymbol{o}_1, \ldots, \boldsymbol{o}_{N+1}) \in \mathbb{R}^{(\tilde{d}+1)\times(N+1)}$ *be outputs and* $\boldsymbol{h}_l^{(i)}$ *'s be hidden states. For each* $i \in [\tilde{d}+1]$ *and* $l \in [N+1]$, *we have*

$$\boldsymbol{h}_l^{(i)} = \sum_{j=1}^{l} G_{j,l}(\boldsymbol{Z})(\boldsymbol{z}_j)_i \boldsymbol{W}_B \boldsymbol{z}_j, \quad \boldsymbol{o}_l[i] = \sum_{j=1}^{l} G_{j,l}(\boldsymbol{Z})(\boldsymbol{z}_j)_i \boldsymbol{z}_j^\top \boldsymbol{W}_B^\top \boldsymbol{W}_C \boldsymbol{z}_l,$$

*where* $G_{j,l}(\boldsymbol{Z}) = \sigma\left(\boldsymbol{w}^\top \boldsymbol{z}_j + b\right) \prod_{k=j+1}^{l}\left(1 - \sigma\left(\boldsymbol{w}^\top \boldsymbol{z}_k + b\right)\right)$.

*Proof of Lemma A.1.* For each $l \in [N+1]$, we have

$$\overline{\boldsymbol{A}}_l = \exp\left(-\mathrm{softplus}(\boldsymbol{w}^\top \boldsymbol{z}_l + b)\boldsymbol{I}_{\tilde{d}+1}\right) = \frac{1}{1 + \exp(\boldsymbol{w}^\top \boldsymbol{z}_l + b)}\boldsymbol{I}_{\tilde{d}+1} = \left(1 - \sigma\left(\boldsymbol{w}^\top \boldsymbol{z}_l + b\right)\right)\boldsymbol{I}_{\tilde{d}+1}$$

and

$$\overline{\boldsymbol{B}}_l = -\left(\overline{\boldsymbol{A}}_l - \boldsymbol{I}_{\tilde{d}+1}\right)\boldsymbol{W}_B \boldsymbol{z}_l = \sigma\left(\boldsymbol{w}^\top \boldsymbol{z}_l + b\right)\boldsymbol{W}_B \boldsymbol{z}_l.$$

We fix any $i \in [\tilde{d}+1]$ and we will prove by applying induction on $l \in [N+1]$. Let us first consider the case $l = 1$. We have

$$\boldsymbol{h}_1^{(i)} = \left(1 - \sigma\left(\boldsymbol{w}^\top \boldsymbol{z}_l + b\right)\right)\boldsymbol{h}_0^{(i)} + \sigma\left(\boldsymbol{w}^\top \boldsymbol{z}_1 + b\right)\boldsymbol{W}_B \boldsymbol{z}_1 (\boldsymbol{z}_1)_i$$
$$= \sigma\left(\boldsymbol{w}^\top \boldsymbol{z}_1 + b\right)\boldsymbol{W}_B \boldsymbol{z}_1 \boldsymbol{z}_1[i]$$
$$= G_{1,1}(\boldsymbol{Z})\boldsymbol{z}_1[i], \boldsymbol{W}_B \boldsymbol{z}_1$$

and

$$\boldsymbol{o}_1[i] = (\boldsymbol{W}_C \boldsymbol{z}_1)^\top \boldsymbol{h}_1^{(i)} = \sigma\left(\boldsymbol{w}^\top \boldsymbol{z}_1 + b\right)\boldsymbol{z}_1[i]\boldsymbol{z}_1^\top \boldsymbol{W}_B^\top \boldsymbol{W}_C \boldsymbol{z}_1 = G_{1,1}(\boldsymbol{Z})\boldsymbol{z}_1[i]\boldsymbol{z}_1^\top \boldsymbol{W}_B^\top \boldsymbol{W}_C \boldsymbol{z}_1.$$

Therefore, desired conclusions hold for the case $l = 1$.

Next, we assume that our conclusion holds for $l < \tilde{d} + 1$.

$$\boldsymbol{h}_{l+1}^{(i)} = \left(1 - \sigma\left(\boldsymbol{w}^\top \boldsymbol{z}_{l+1} + b\right)\right)\boldsymbol{h}_l^{(i)} + \sigma\left(\boldsymbol{w}^\top \boldsymbol{z}_{l+1} + b\right)\boldsymbol{W}_B \boldsymbol{z}_{l+1}(\boldsymbol{z}_{l+1})_i$$

$$= \left(1 - \sigma\left(\boldsymbol{w}^\top \boldsymbol{z}_{l+1} + b\right)\right)\sum_{j=1}^{l} G_{j,l}(\boldsymbol{Z})\boldsymbol{z}_j[i]\boldsymbol{W}_B \boldsymbol{z}_j + G_{l+1,l+1}(\boldsymbol{Z})\boldsymbol{z}_{l+1}[i]\boldsymbol{W}_B \boldsymbol{z}_{l+1}$$

$$= \sum_{j=1}^{l+1} G_{j,l+1}(\boldsymbol{Z})\boldsymbol{z}_j[i]\boldsymbol{W}_B \boldsymbol{z}_j$$

and

$$\boldsymbol{o}_{l+1}[i] = (\boldsymbol{W}_C \boldsymbol{z}_{l+1})^\top \boldsymbol{h}_{l+1}^{(i)}$$

$$= (\boldsymbol{W}_C \boldsymbol{z}_{l+1})^\top \sum_{j=1}^{l+1} G_{j,l+1}(\boldsymbol{Z})\boldsymbol{z}_j[i]\boldsymbol{W}_B \boldsymbol{z}_j$$

$$= (\boldsymbol{o}_{l+1})_i = \sum_{j=1}^{l+1} G_{j,l+1}(\boldsymbol{Z})\boldsymbol{z}_j[i]\boldsymbol{z}_j^\top \boldsymbol{W}_B^\top \boldsymbol{W}_C \boldsymbol{z}_{l+1}.$$

Therefore, we have the desired conclusions. $\qquad\square$

## A.2 HIGH PROBABILITY EVENTS

Throughout the proof, we use the term "with high probability" which is defined as follows.

**Definition A.2.** We call that an event $E$ occurs *with high probability*, when

$$\mathbb{P}[E] \geq 1 - d^{-C_{\mathrm{whp}}}$$

with a large enough $C_{\mathrm{whp}} > 0$.

For example, $\mathbf{z} = \mathcal{O}(\sqrt{\log d})$ for $\mathbf{z} \sim \mathcal{N}(0,1)$, with high probability, which is a direct consequence of Hoeffding's inequality. In addition, the intersection of a $\mathrm{poly}(d)$ events also occurs with high probability. We use these property frequently throughout our proof.

The following lemma is useful when we bound some quantities with high probability.

**Lemma A.3** (Corollary 17 in Oko et al. (2024), adapted). *Let $P$ be a polynomial with degree $\deg(P)$. If $\|\boldsymbol{\beta}\| = 1$ and $\boldsymbol{x} \sim \mathcal{N}(\mathbf{0}, \boldsymbol{I}_d)$, then $|P(\langle \boldsymbol{\beta}, \boldsymbol{x} \rangle)| \lesssim (\log d)^{\deg(P)/2}$ holds with high probability.*

This lemma implies that $y_i^t = \tilde{\mathcal{O}}(1), \|\phi(\boldsymbol{x}_i^t)\|, \|\phi(\boldsymbol{x}^t)\| = \tilde{\mathcal{O}}(d)$ holds for any $i \in [N_{\mathrm{pt}}], t \in [T_{\mathrm{pt}}]$, with high probability. We utilize these properties frequently in our proof.

The following lemma provides a high-probability guarantee regarding our input embedding, which is crucial for our analysis.

**Lemma A.4.** *Let $\mathbf{x}_1, \ldots, \mathbf{x}_N \sim \mathcal{N}(0,1)$ and let $\mathbf{z}_1, \ldots, \mathbf{z}_N$ be i.i.d. random variables such that $|\mathbf{z}_i| \leq C$ with high probability where $\mathbf{z}_i$ might depend on $\mathbf{x}_i$. If $N = \tilde{\Omega}(C^2)$ and $N \leq N^*$, then for each $k = 0, 1, 2$,*

$$\left| \frac{1}{N} \sum_{i=1}^{N} \mathbf{z}_i \mathrm{He}_k(\mathbf{x}_i) - \mathbb{E}[\mathbf{z}_1 \mathrm{He}_k(\mathbf{x}_1)] \right| \leq \tilde{\mathcal{O}}\left( CN^{-1/2} \right),$$

*with high probability. In addition, let $\boldsymbol{x}_1', \ldots, \boldsymbol{x}_N' \sim \mathcal{N}(0,1)$, then under the same condition,*

$$\left| \frac{1}{N} \sum_{i=1}^{N} \mathbf{z}_i \mathbf{x}_i \mathbf{x}_i' - \mathbb{E}\left[ \mathbf{z}_1 \mathbf{x}_i \mathbf{x}_i' \right] \right| \leq \tilde{\mathcal{O}}\left( CN^{-1/2} \right),$$

*with high probability.*

*Proof of Lemma A.4.* Let $\mathbf{z}_i' := \mathbb{1}[|\mathbf{z}_i| \leq C]\mathbf{z}_i$. Then, $\mathbf{z}_i', \mathbf{z}_i'\mathbf{x}_i, \mathbf{z}_i'\mathrm{He}_2(\mathbf{x}_i), \mathbf{z}_i'\mathbf{x}_i\mathbf{x}_i'$ are $C$-subexponential. Since $N = \tilde{\Omega}(C^2)$, for each $k = 0, 1, 2$, we have

$$\left| \frac{1}{N} \sum_{i=1}^{N} \mathbf{z}_i' \mathrm{He}_k(\mathbf{x}_i) - \mathbb{E}[\mathbf{z}_1' \mathrm{He}_k(\mathbf{x}_i)] \right| \leq \tilde{\mathcal{O}}\left( CN^{-1/2} \right),$$

with probability and $\sum_{i=1}^{N} \mathbf{z}_i' \mathrm{He}_k(\mathbf{x}_i) = \sum_{i=1}^{N} \mathbf{z}_i \mathrm{He}_k(\mathbf{x}_1)$ with high probability. In addition, we have

$$|\mathbb{E}[\mathbf{z}_1 \mathrm{He}_k(\mathbf{x}_i)] - \mathbb{E}[\mathbf{z}' \mathrm{He}_k(\mathbf{x}_i)]| = \mathbb{E}[\mathbb{1}[|\mathbf{z}_1| \geq C]\mathrm{He}_k(\mathbf{x}_i)]$$

$$\leq \mathbb{P}[\mathbb{1}[|\mathbf{z}_1| \geq C]\mathbb{E}\left[ \left( \mathrm{He}_k(\mathbf{x}_1) \right)^2 \right]^{\frac{1}{2}}$$

$$\leq \frac{1}{\mathrm{poly}(d)}.$$

Therefore, by combining the two bounds above, we have the desired conclusion for the case $k = 0, 1, 2$. Using the same argument, we can also obtain the last conclusion. $\qquad\square$

## A.3 REDUCING THE INFORMATION EXPONENT WITH LABEL TRANSFORMATION

For any function $h$ which is $L^2$ integrable with respect to Gaussian measure and $p \in \mathbb{N} \cup \{0\}$, we define $e_p(h) := \min\{i \in \mathbb{N} : H(f^i, p) \neq 0\}$. If this minimum does not exist (i.e., the set is empty), we set $e_p(f) = \infty$. In addition, we define

$$\bar{g}_*(z) := \begin{cases} \frac{g_*(z)}{\rho} & \text{if } \left| \frac{g_*(z)}{\rho} \right| \leq \frac{1}{\log d} \\ 0 & \text{otherwise} \end{cases}.$$

From our choice of $\rho$, $\bar{g}_*\left(y_i^t\right) = g_*\left(y_i^t\right)/\rho$ for all $i \in [N_{\mathrm{pt}}], t \in [T_{\mathrm{pt}}]$, with high probability. We use this frequently in our proof. We also define the following function, which naturally appears in our analysis:

$$A(z) := \frac{1}{2}\left[(\rho\bar{g}_*(z) + \tau)\sigma\left(\bar{g}_*(z) + \tau/\rho - b\right) + (\rho\bar{g}_*(z) - \tau)\sigma\left(\bar{g}_*(z) - \tau/\rho - b\right)\right].$$

Let $A(z) = \sum_{k=0}^{\infty} \frac{a_k}{k!}\mathrm{He}_k(z)$ be the Hermite expansion. The following lemma characterizes its Hermite coefficients.

**Lemma A.5.** *For any $p \in \mathbb{N} \cup \{0\}$, if $e_p(g_*) < \infty$, we have*

$$d^{C_b}a_p = \Theta\left((\log d)^{-C_\rho(e_p(g_*)-1)}\right),$$

*where hidden constants depend on $g_*$ and $p$. In addition, if $e_p(g_*) = \infty$, then $d^{C_b}|a_p| \lesssim 1/\mathrm{poly}(d)$.*

The following two lemmas are crucial for our proof of Lemma A.5.

**Lemma A.6** (Proposition 6 in Lee et al. (2024))**.** *For any polynomial $P$, there exist $C_P, D_P > 0$ depending only on $P$ such that the following holds.*

- *If $P$ is not an even function, then there exists $i \le C_P$ such that $\left|H(f^i, 1)\right| \ge D_P$.*

- *If $P$ is an even function, then there exists $i \le C_P$ such that $\left|H(f^i, 2)\right| \ge D_P$.*

**Lemma A.7.** *For any $k \in \mathbb{N} \cup \{0\}$ and $z < -k - 2$, we have $\frac{e^z}{2} \le \sigma^{(k)}(z) \le 2e^z$.*

*Proof of Lemma A.7.* For any $x < 0$, we have

$$\sigma(x) = \frac{1}{1 + \exp(-x)} = 1 - \frac{1}{1 + \exp(x)} = \sum_{j=1}^{\infty}(-1)^{j-1}e^{jx}.$$

Therefore, we have

$$\sigma^{(k)}(z) = \sum_{j=1}^{\infty}(-1)^{j-1}j^k e^{jz} = e^z + \sum_{j=2}^{\infty}(-1)^{j-1}j^k e^{jz}.$$

For each $j \ge 2$, since $\frac{(j+1)^k e^{(j+1)z}}{j^k e^{jz}} \le 2^k e^z$ and $2^k e^z < \frac{1}{3}$, we have

$$\left|\sigma^{(k)}(z) - e^z\right| \le \sum_{j=2}^{\infty}j^k e^{jz} \le 2^k e^{2z}\sum_{j=0}^{\infty}\left(2^k e^z\right)^j = \frac{2^k e^{2z}}{1 - 2^k e^z} \le \frac{e^z}{2}.$$

Hence, we have a desired conclusion. $\square$

We now prove Lemma A.5.

*Proof of Lemma A.5.* By applying Taylor's theorem for $\sigma(\cdot)$ at points $\pm\tau/\rho + b$, for any $z \in \mathbb{R}$, we have

$$2\rho^{-1}A(z)$$
$$= (\bar{g}_*(z) + \tau/\rho)\sigma\left(\bar{g}_*(z) + \tau/\rho - b\right) + (\bar{g}_*(z) - \tau/\rho)\sigma\left(\bar{g}_*(z) - \tau/\rho - b\right)$$
$$= \sum_{i=0}^{e_p(g_*)-1}(s_i + \tilde{s}_i)\bar{g}_*^{i+1}(z) + \left(R(z) + \tilde{R}(z)\right)\bar{g}_*^{e_p(g_*)+1}(z)$$
$$+ \tau\left[\sum_{i=0}^{e_p(g_*)-1}(s_i - \tilde{s}_i)\bar{g}_*^i(z) + \left(R(z) - \tilde{R}(z)\right)\bar{g}_*^{e_p(g_*)}(z)\right]$$

where $s_i = \frac{\sigma^{(i)}(\tau/\rho+b)}{i!}, \tilde{s}_i = \frac{\sigma^{(i)}(-\tau/\rho+b)}{i!}$ for $i = 0, \ldots, e_p(g_*) - 1$ and

$$|R(z)|, |\tilde{R}(z)| \le \frac{\max_{t \in [b-1,b+1]}\left|\sigma^{(e_p(g_*))}(t)\right|}{(e_p(g_*))!} \le \frac{2e^{b+1}}{(e_p(g_*))!}.$$

From the definition of $\bar{g}_*$, we have

$$\sum_{i=0}^{e_p(g_*)-1} (s_i + \tilde{s}_i)\bar{g}_*^{i+1}(z) + \tau \sum_{i=0}^{e_p(g_*)-1} (s_i - \tilde{s}_i)\bar{g}_*^i(z)$$

$$= \sum_{i=0}^{e_p(g_*)-1} \rho^{-(i+1)}(s_i + \tilde{s}_i)g_*^{i+1}(z) + \tau \sum_{i=0}^{e_p(g_*)-1} \rho^{-i}(s_i - \tilde{s}_i)g_*^i(z)$$

$$- \left( \sum_{i=0}^{e_p(g_*)-1} \rho^{-(i+1)}(s_i + \tilde{s}_i)g_*^{i+1}(z) + \tau \sum_{i=0}^{e_p(g_*)-1} \rho^{-i}(s_i - \tilde{s}_i)g_*^i(z) \right) \mathbb{1}\left[ \left| \frac{g_*(z)}{\rho} \right| \leq \frac{1}{\log d} \right].$$

For any $i = 0, \ldots, e_p(g_*)$, we have

$$\left| \mathbb{E}_{\mathbf{z} \sim \mathcal{N}(0,1)} \left[ g_*^i(\mathbf{z}) \mathbb{1}\left[ \left| \frac{g_*(z)}{\rho} \right| \leq \frac{1}{\log d} \right] \right] \right| \leq \mathbb{E}_{\mathbf{z} \sim \mathcal{N}(0,1)} \left[ g_*^{2i}(\mathbf{z}) \right]^{\frac{1}{2}} \mathbb{P}_{\mathbf{z} \sim \mathcal{N}(0,1)} \left[ \left| \frac{g_*(\mathbf{z})}{\rho} \right| \leq \frac{1}{\log d} \right]$$

$$= o\left( \frac{1}{\text{polylog}(d)} \right).$$

Combining this with additivity of $H(\cdot, p)$ and the fact that $\mathbb{E}_{\mathbf{z} \sim \mathcal{N}(0,1)}[g_*(\mathbf{z})] = 0$, we have

$$2\rho^{-1}a_p = \rho^{-e_p(g_*)}(s_{e_p(g_*)-1} + \tilde{s}_{e_p(g_*)-1})H\left( g_*^{e_p(g_*)}, p \right)$$

$$+ H\left( (R + \tilde{R}) \cdot \bar{g}_*^{e_p(g_*)+1}, p \right) + \tau H\left( (R - \tilde{R}) \cdot \bar{g}_*^{e_p(g_*)}, p \right) \pm o\left( \frac{1}{\text{polylog}(d)} \right).$$

From our choice of $b$ and $\rho$, the first term is $\Theta(d^{-C_b}(\log d)^{-C_\rho e_p(g_*)})$. Next, we bound the second term. For any $z \in \mathbb{R}$, we have

$$\left| H\left( (R + \tilde{R}) \cdot \bar{g}_*^{e_p(g_*)+1}, p \right) \right|$$

$$= \left| \mathbb{E}_{\mathbf{z} \sim \mathcal{N}(0,1)} \left[ \text{He}_p(\mathbf{z})(R(\mathbf{z}) + \tilde{R}(\mathbf{z}))\bar{g}_*^{e_p(g_*)+1}(\mathbf{z}) \right] \right|$$

$$\leq \mathbb{E}_{\mathbf{z} \sim \mathcal{N}(0,1)} \left[ \left| \text{He}_p(\mathbf{z})(R(\mathbf{z}) + \tilde{R}(\mathbf{z}))\bar{g}_*^{e_p(g_*)+1}(\mathbf{z}) \right| \right]$$

$$\leq \frac{2e^{b+1}}{(e_p(g_*)+1)!} \mathbb{E}_{\mathbf{z} \sim \mathcal{N}(0,1)} \left[ \text{He}_p(\mathbf{z})^2 \right]^{\frac{1}{2}} \mathbb{E}_{\mathbf{z} \sim \mathcal{N}(0,1)} \left[ \bar{g}_*(\mathbf{z})^{2e_p(g_*)+2} \right]^{\frac{1}{2}}$$

$$\leq \frac{2e^{b+1}\rho^{-(e_p(g_*)+1)}}{(e_p(g_*)+1)!} \mathbb{E}_{\mathbf{z} \sim \mathcal{N}(0,1)} \left[ \text{He}_p(\mathbf{z})^2 \right]^{\frac{1}{2}} \mathbb{E}_{\mathbf{z} \sim \mathcal{N}(0,1)} \left[ g_*(\mathbf{z})^{2e_p(g_*)+2} \right]^{\frac{1}{2}},$$

where we apply the Cauchy–Schwarz inequality for the second inequality. Hence, the absolute value of the second term is $\tilde{\mathcal{O}}(d^{-C_b}(\log d)^{-C_\rho(e_p(g_*)+1)})$. Using a similar argument, we can know that the absolute value of the third term is $\Theta\left( d^{-C_b}(\log d)^{-C_\rho e_p(g_*)} \right)$. Combining with the fact that $\tau$ is small enough, we have our desired conclusion.

Using similar arguments, we can also obtain our conclusion for the case $e_p(g_*) = \infty$. $\qquad\square$

## B ONE-STEP GRADIENT DESCENT ON THE MAMBA LAYER

Let us define the function $\psi : \mathbb{R}^{d+3} \to \mathbb{R}^{\tilde{d}}$, which we use repeatedly in our proof. It is defined as

$$\psi(\boldsymbol{\theta}, c_0, c_1, c_2) := \left[ c_0, c_1\boldsymbol{\theta}^\top, \frac{c_2(\boldsymbol{\theta} \odot \boldsymbol{\theta})^\top}{\sqrt{2}}, c_2\boldsymbol{\theta}[1]\boldsymbol{\theta}[2], \ldots, c_2\boldsymbol{\theta}[d-1]\boldsymbol{\theta}[d] \right]^\top.$$

Note that for any vector $\boldsymbol{\theta} \in \mathbb{R}^d$,

$$\|\psi(\boldsymbol{\theta}, c_0, c_1, c_2)\|^2 = c_0^2 + c_1^2 \|\boldsymbol{\theta}\|^2 + \frac{c_2^2 \|\boldsymbol{\theta}\|^4}{2}. \tag{3}$$

If we choose $\lambda_1 = \eta^{-1}$, the updated parameter $\boldsymbol{\gamma}^*$ can be expressed as

$$\boldsymbol{\gamma}^* = \frac{2\eta}{T_1} \sum_{t \in [T_1]} \left[ \left( y^t - f\left(\boldsymbol{Z}^t; \boldsymbol{\gamma}(0), \boldsymbol{u}(0), \boldsymbol{v}(0), \boldsymbol{a}(0)\right) \right) \nabla_{\boldsymbol{\gamma}} f\left(\boldsymbol{Z}^t; \boldsymbol{\gamma}(0), \boldsymbol{u}(0), \boldsymbol{v}(0), \boldsymbol{a}(0)\right) \right]$$

$$= \frac{2\eta}{T_1} \sum_{t \in [T_1]} y^t \nabla_{\boldsymbol{\gamma}} f\left(\boldsymbol{Z}^t; \boldsymbol{\gamma}(0), \boldsymbol{u}(0), \boldsymbol{v}(0), \boldsymbol{a}(0)\right)$$

$$- \frac{2\eta}{T_1} \sum_{t \in [T_1]} f\left(\boldsymbol{Z}^t; \boldsymbol{\gamma}(0), \boldsymbol{u}(0), \boldsymbol{v}(0), \boldsymbol{a}(0)\right) \Big) \nabla_{\boldsymbol{\gamma}} f\left(\boldsymbol{Z}^t; \boldsymbol{\gamma}(0), \boldsymbol{u}(0), \boldsymbol{v}(0), \boldsymbol{a}(0)\right).$$

The initial output evaluated at $\boldsymbol{Z}^t$ can be bounded as

$$\left| f\left(\boldsymbol{Z}^t; \boldsymbol{\gamma}(0), \boldsymbol{u}(0), \boldsymbol{v}(0), \boldsymbol{a}(0)\right) \right|$$

$$= \mathrm{ReLU}\left( N_{\mathrm{pt}}^{-1} \sum_{j=1}^{N_{\mathrm{pt}}} G_{j,n+1}(\boldsymbol{Z}) y_j^t \phi\left(\boldsymbol{x}_j^t\right)^\top \left(\boldsymbol{\gamma}(0) \odot \phi\left(\boldsymbol{x}^t\right)\right) \right)$$

$$\leq N_{\mathrm{pt}}^{-1} \sum_{h=1}^{N_{\mathrm{pt}}} G_{j,n+1}(\boldsymbol{Z}) \left| y_j^t \right| \left\| \phi\left(\boldsymbol{x}_j^t\right) \right\| \left\| \phi\left(\boldsymbol{x}^t\right) \right\|$$

$$= \tilde{\mathcal{O}}\left( d^{-C_b+2} \right),$$

with high probability.

The gradient of $\boldsymbol{\gamma}$ of output evaluated at $\boldsymbol{Z}^t$ can be calculated as

$$\nabla_{\boldsymbol{\gamma}} f\left(\boldsymbol{Z}^t; \boldsymbol{\gamma}(0), \boldsymbol{u}(0), \boldsymbol{v}(0), \boldsymbol{a}(0)\right)$$

$$= \mathbb{1}\left[ \sum_{j=1}^{N_{\mathrm{pt}}} G_{j,N_{\mathrm{pt}}+1}\left(\boldsymbol{Z}^t\right) y_j^t \phi\left(\boldsymbol{x}_j^t\right)^\top \left(\boldsymbol{\gamma}(0) \odot \phi\left(\boldsymbol{x}^t\right)\right) > 0 \right]$$

$$\times \left( N_{\mathrm{pt}}^{-1} \sum_{j=1}^{N_{\mathrm{pt}}} G_{j,N_{\mathrm{pt}}+1}\left(\boldsymbol{Z}^t\right) y_j^t \phi\left(\boldsymbol{x}_j^t\right) \odot \phi\left(\boldsymbol{x}^t\right) \right)$$

and its norm can be bounded as

$$\left\| \nabla_{\boldsymbol{\gamma}} f\left(\boldsymbol{Z}^t; \boldsymbol{\gamma}(0), \boldsymbol{u}(0), \boldsymbol{v}(0), \boldsymbol{a}(0)\right) \right\| \leq N_{\mathrm{pt}}^{-1} \sum_{h=1}^{N_{\mathrm{pt}}} G_{j,n+1}(\boldsymbol{Z}) \left| y_j^t \right| \left\| \phi\left(\boldsymbol{x}_j^t\right) \right\| \left\| \phi\left(\boldsymbol{x}^t\right) \right\|$$

$$= \tilde{\mathcal{O}}\left( d^{-C_b+2} \right),$$

with high probability. Therefore, with high probability, we have

$$\boldsymbol{\gamma}^* = \frac{2\eta}{T_1} \sum_{t \in [T_1]} y^t \nabla_{\boldsymbol{\gamma}} f\left(\boldsymbol{Z}^t; \boldsymbol{\gamma}(0), \boldsymbol{u}(0), \boldsymbol{v}(0), \boldsymbol{a}(0)\right) + \tilde{\mathcal{O}}\left( \eta d^{-2C_b+4} \right).$$

Hence, we will focus on estimating the first term.

## B.1 ESTIMATION OF LABEL-GRADIENT CORRELATION

We first establish a high-probability guarantee for the term containing context examples.

**Lemma B.1.** *Let $(\boldsymbol{x}_1, y_1, \ldots, \boldsymbol{x}_N, y_N, \boldsymbol{x})$ be a prompt with context length $N \leq N^*$ and feature vector $\boldsymbol{\beta} \in \mathbb{R}^d$ and its embedding $\boldsymbol{Z} \in \mathbb{R}^{(\tilde{d}+1) \times (N+1)}$. Then, the following holds with high probability:*

$$N^{-1} \sum_{j=1}^{N} G_{j,N+1}\left(\boldsymbol{Z}\right) y_j \phi\left(\boldsymbol{x}_j\right) = \psi\left(\boldsymbol{\beta}, a_0, a_1, a_2\right) + \tilde{\mathcal{O}}\left( d^{-C_b+1} N^{-1/2} \right).$$

*Proof of Lemma B.1.* Note that with high probability, $y_j/\rho = \bar{g}_* (\langle \boldsymbol{\beta}, \boldsymbol{x}_j \rangle) + \zeta_j/\rho$ with $\zeta_j \sim$ Unif$(\{-\tau, \tau\})$ for all $j \in [N]$. Condition on this event, we have

$$N^{-1} \sum_{j=1}^{N} y_j \sigma(y_j/\rho + b) \phi(\boldsymbol{x}_j)$$

$$= N^{-1} \sum_{j=1}^{N} \left[ (\rho \bar{g}_* (\langle \boldsymbol{\beta}, \boldsymbol{x}_j \rangle) + \zeta_j) \sigma (\bar{g}_* (\langle \boldsymbol{\beta}, \boldsymbol{x}_j \rangle) + \zeta_j/\rho + b) \phi(\boldsymbol{x}_j) \right].$$

From Stein's lemma, we have

$$\mathbb{E} \left[ N^{-1} \sum_{j=1}^{N} \left[ (\rho \bar{g}_* (\langle \boldsymbol{\beta}, \boldsymbol{x}_j \rangle) + \zeta_j) \sigma (\bar{g}_* (\langle \boldsymbol{\beta}, \boldsymbol{x}_j \rangle) + \zeta_j/\rho + b) \phi(\boldsymbol{x}_j^t) \right] \right]$$

$$= \mathbb{E}_{\boldsymbol{x} \sim \mathcal{N}(\boldsymbol{0}, \boldsymbol{I}_d)} \left[ A \left( \langle \boldsymbol{\beta}^t, \boldsymbol{x} \rangle \right) \phi(\boldsymbol{x}) \right]$$

$$= \psi \left( \boldsymbol{\beta}^t, a_0, a_1, a_2 \right).$$

By Lemma A.4, with high probability, we have

$$\left\| N^{-1} \sum_{j=1}^{N} y_j \sigma(y_j/\rho + b) \phi(\boldsymbol{x}_j) - \psi (\boldsymbol{\beta}, a_0, a_1, a_2) \right\| \leq \tilde{\mathcal{O}} \left( d^{-C_b+1} N^{-1/2} \right).$$

In addition, with high probability, we have

$$\left\| N^{-1} \sum_{j=1}^{N} G_{j,n+1} (\boldsymbol{Z}) y_j \phi(\boldsymbol{x}_j) - N^{-1} \sum_{j=1}^{N} y_j \sigma(y_j/\rho + b) \phi(\boldsymbol{x}_j) \right\|$$

$$= \left\| N^{-1} \sum_{j=1}^{N} \left[ y_j \sigma(y_j/\rho + b) \left( 1 - (1 - \sigma(b)) \prod_{i=j+1}^{N} (1 - \sigma(y_j/\rho + b)) \right) \phi(\boldsymbol{x}_j) \right] \right\|$$

$$\leq N^{-1} \sum_{j=1}^{N} \left[ |y_j \sigma(y_i/\rho + b)| \left( 1 - (1 - \sigma(b)) \prod_{i=j+1}^{N} (1 - \sigma(y_i/\rho + b)) \right) \|\phi(\boldsymbol{x}_j)\| \right]$$

$$\leq N^{-1} \sum_{j=1}^{N} \left[ |y_j \sigma(y_j/\rho + b)| \left( 1 - (1 - \sigma(2b))^{N^*} \right) \|\phi(\boldsymbol{x}_j)\| \right]$$

$$\leq \tilde{\mathcal{O}}(d^{-2C_b+C^*}).$$

From a large enough choice of $C_b$ and the triangular inequality, we have the desired conclusion. $\square$

**Corollary B.2.** *For each $t \in [T_1]$, the following holds with high probability:*

$$N_{\text{pt}}^{-1} \sum_{j=1}^{N_{\text{pt}}} G_{j,N_{\text{pt}}+1} \left( \boldsymbol{Z}^t \right) y_j^t \phi \left( \boldsymbol{x}_j^t \right)^\top \left( \boldsymbol{\gamma}(0) \odot \phi \left( \boldsymbol{x}^t \right) \right)$$

$$= a_0 \gamma^2 + a_1 \langle \boldsymbol{\beta}^t, \boldsymbol{x}^t \rangle + a_2 \gamma \text{He}_2 \left( \langle \boldsymbol{\beta}^t, \boldsymbol{x}^t \rangle \right) + \tilde{\mathcal{O}} \left( d^{-C_b+2} N_{\text{pt}}^{-1/2} \right).$$

*Proof of Corollary B.2.* From Lemma B.1, for each $t \in [T_1]$, with high probability, we have

$$N_{\text{pt}}^{-1} \sum_{j=1}^{N_{\text{pt}}} G_{j,N_{\text{pt}}+1} \left( \boldsymbol{Z}^t \right) y_j^t \phi \left( \boldsymbol{x}_j^t \right)^\top \left( \boldsymbol{\gamma}(0) \odot \phi \left( \boldsymbol{x}^t \right) \right)$$

$$= a_0 \gamma^2 + a_1 \langle \boldsymbol{\beta}^t, \boldsymbol{x}^t \rangle + a_2 \gamma \text{He}_2 \left( \langle \boldsymbol{\beta}^t, \boldsymbol{x}^t \rangle \right) + \tilde{\mathcal{O}} \left( d^{-C_b+1} N_{\text{pt}}^{-1/2} \right) \|\phi \left( \boldsymbol{x}^t \right)\|$$

$$= a_0 \gamma^2 + a_1 \langle \boldsymbol{\beta}^t, \boldsymbol{x}^t \rangle + a_2 \gamma \text{He}_2 \left( \langle \boldsymbol{\beta}^t, \boldsymbol{x}^t \rangle \right) + \tilde{\mathcal{O}} \left( d^{-C_b+2} N_{\text{pt}}^{-1/2} \right),$$

where we use Lemma A.3 and (3) for the last equality. $\square$

Next, let us estimate the expectation of the first term that appears in the label-gradient correlation. The following lemma is useful for this purpose.

**Lemma B.3.** *For any $\delta > 0$ with $\delta = \tilde{\mathcal{O}}(d^{-C_\delta})$ for some constant $C > 0$, the following holds:*

$$\mathbb{P}_{\mathbf{z}\sim\mathcal{N}(0,1)}\left[\left|a_0\gamma^2 + a_1\mathbf{z} + a_2\gamma\mathrm{He}_2\left(\mathbf{z}\right)\right| < d^{-C_b}\delta\right] \leq \tilde{\mathcal{O}}\left(d^{-C_\delta}\right).$$

*Proof of Lemma B.3.* For simplicity, let $\delta' = d^{-C_b}\delta$. Note that $e_0(g_*) = 2$. Then, from Lemma A.5, we know that $d^{C_b}a_0 = \Theta\left((\log d)^{-2C_\rho}\right)$. In addition, for $p = 1, 2$, $d^{C_b}a_p = \Theta\left((\log d)^{-C_\rho e_p(g_*)}\right)$ if $e_p(g_*) < \infty$ and $d^{C_b}a_p \lesssim \frac{1}{\mathrm{poly}(d)}$ otherwise.

**Case 1:** $a_2 = 0$.

In this case, $g_*$ is not an even function and then $d^{C_b}a_1, d^{C_b}a_0 = \tilde{\Theta}(1)$. Without loss of generality, we assume $a_1 > 0$. Then, we have

$$\mathbb{P}_{\mathbf{z}\sim\mathcal{N}(0,1)}\left[\left|a_0\gamma^2 + a_1\mathbf{z} + a_2\gamma\mathrm{He}_2\left(\mathbf{z}\right)\right| < d^{-C_b}\delta\right]$$
$$= \mathbb{P}_{\mathbf{z}\sim\mathcal{N}(0,1)}\left[-\left(\delta' + a_0\gamma^2 - a2\gamma\right)/a_1 < \mathbf{z} < \left(\delta' - a_0\gamma^2 + a_2\gamma\right)/a_1\right]$$
$$\leq \frac{2\delta'}{\sqrt{2\pi}a_1} = \tilde{\mathcal{O}}\left(d^{-C_\delta}\right).$$

**Case 2:** $a_2 \neq 0$.

Without loss of generality, we assume $a_2 > 0$. Then, we have

$$\mathbb{P}_{\mathbf{z}\sim\mathcal{N}(0,1)}\left[\left|a_0\gamma^2 + a_1\mathbf{z} + a_2\gamma\mathrm{He}_2(\mathbf{z})\right| < d^{-C_b}\delta\right]$$
$$= \mathbb{P}_{\mathbf{z}\sim\mathcal{N}(0,1)}\left[\frac{-a_1 - \sqrt{a_1^2 + 4a_2\gamma(a_2\gamma - a_0\gamma^2 + \delta')}}{2a_2\gamma} < \mathbf{z}\right]$$
$$- \mathbb{P}_{\mathbf{z}\sim\mathcal{N}(0,z)}\left[\frac{-a_1 - \sqrt{a_1^2 + 4a_2\gamma(a_2\gamma - a_0\gamma^2 - \delta')}}{2a_2\gamma} < \mathbf{z}\right]$$
$$+ \mathbb{P}_{\mathbf{z}\sim\mathcal{N}(0,1)}\left[\frac{-a_1 + \sqrt{a_1^2 + 4a_2\gamma(a_2\gamma - a_0\gamma^2 - \delta')}}{2a_2\gamma} < \mathbf{z}\right]$$
$$- \mathbb{P}_{\mathbf{z}\sim\mathcal{N}(0,z)}\left[\frac{-a_1 + \sqrt{a_1^2 + 4a_2\gamma(a_2\gamma - a_0\gamma^2 + \delta')}}{2a_2\gamma} < \mathbf{z}\right]$$
$$\leq \frac{\sqrt{a_1^2 + 4a_2\gamma(a_2\gamma - a_0\gamma^2 + \delta')} - \sqrt{a_1^2 + 4a_2\gamma(a_2\gamma - a_0\gamma^2 - \delta')}}{\sqrt{2\pi}a_2\gamma}$$
$$= \frac{4\delta'}{\sqrt{2\pi}\left(\sqrt{a_1^2 + 4a_2\gamma(a_2\gamma - a_0\gamma^2 + \delta')} + \sqrt{a_1^2 + 4a_2\gamma(a_2\gamma - a_0\gamma^2 - \delta')}\right)}.$$

For the case $g_*$ is not an even function, then $d^{C_b}a_0, d^{C_b}a_1 = \tilde{\Theta}(1)$ and $d^{C_b}|a_2| \lesssim 1/\mathrm{poly}(d)$. If $g_*$ is an even function, then $d^{C_b}a_0, d^{C_b}a_2 = \tilde{\Theta}(1)$ and $d^{C_b}a_1 \lesssim 1/\mathrm{poly}(d)$. In both cases, we can check that the term above is $\tilde{\mathcal{O}}(d^{-C_\delta})$. $\square$

For each $t \in [T_1]$, define an event $E_t$ such that

$$\mathbb{1}\left[N_{\mathrm{pt}}^{-1}\sum_{j=1}^{N_{\mathrm{pt}}} G_{j,N_{\mathrm{pt}}+1}\left(\boldsymbol{Z}^t\right)y_j^t\phi\left(\boldsymbol{x}_j^t\right)^\top\boldsymbol{\gamma}(0)\phi\left(\boldsymbol{x}^t\right) > 0\right]$$
$$\neq \mathbb{1}\left[a_0\gamma^2 + a_1\left\langle\boldsymbol{\beta}^t\boldsymbol{x}^t\right\rangle + a_2\gamma\mathrm{He}_2\left(\langle\boldsymbol{\beta}^t, \boldsymbol{x}^t\rangle\right) > 0\right].$$

From Corollary B.2 and Lemma B.3, we have

$$\mathbb{P}_{\boldsymbol{Z}^t}[E_t] \leq \mathbb{P}_{\boldsymbol{Z}^t}\left[\left|a_0\gamma^2 + a_1\left\langle\boldsymbol{\beta}^t, \boldsymbol{x}^t\right\rangle + a_2\gamma\mathrm{He}_2\left(\langle\boldsymbol{\beta}^t, \boldsymbol{x}^t\rangle\right)\right| < \tilde{\mathcal{O}}\left(d^{-C_b+2}N_{\mathrm{pt}}^{-1/2}\right)\right] + \frac{1}{\mathrm{poly}(d)}$$

$$= \tilde{\mathcal{O}}\left(d^2 N_{\text{pt}}^{-1/2}\right).$$

Combining with Corollary B.2, with probability at least $1 - \tilde{\mathcal{O}}\left(d^2 T_1 N_{\text{pt}}^{-1/2}\right)$ the following holds: For any $t \in [T_1]$, we have

$$y^t \nabla_{\gamma} f\left(\boldsymbol{Z}^t; \boldsymbol{\gamma}(0), \boldsymbol{u}(0), \boldsymbol{v}(0), \boldsymbol{a}(0)\right)$$

$$= y^t \mathbb{1}\left[N_{\text{pt}}^{-1} \sum_{j=1}^{N_{\text{pt}}} G_{j,N_{\text{pt}}+1}\left(\boldsymbol{Z}^t\right) y_j^t \phi\left(\boldsymbol{x}_j^t\right)^{\top} \left(\boldsymbol{\gamma}(0) \odot \phi\left(\boldsymbol{x}^t\right)\right) > 0\right]$$

$$\times \left(N_{\text{pt}}^{-1} \sum_{j=1} G_{j,N_{\text{pt}}+1}\left(\boldsymbol{Z}^t\right) y_j^t \phi\left(\boldsymbol{x}_j^t\right) \odot \phi\left(\boldsymbol{x}^t\right)\right)$$

$$= y^t \mathbb{1}\left[a_0 \gamma^2 + a_1 \left\langle \boldsymbol{\beta}^t, \boldsymbol{x}^t \right\rangle + a_2 \gamma \text{He}_2\left(\left\langle \boldsymbol{\beta}^t, \boldsymbol{x}^t \right\rangle\right) > 0\right] \phi(\boldsymbol{x}^t) \odot \psi\left(\boldsymbol{\beta}^t, a_0, a_1, a_2\right)$$

$$+ \boldsymbol{n}\left(\boldsymbol{Z}^t\right) y^t \mathbb{1}\left[a_0 \gamma^2 + a_1 \left\langle \boldsymbol{\beta}^t, \boldsymbol{x}^t \right\rangle + a_2 \gamma \text{He}_2\left(\left\langle \boldsymbol{\beta}^t, \boldsymbol{x}^t \right\rangle\right) > 0\right] \phi(\boldsymbol{x}^t) \odot \psi\left(\boldsymbol{\beta}^t, a_0, a_1, a_2\right),$$

where $\|\boldsymbol{n}(\boldsymbol{Z}^t)\| = \tilde{\mathcal{O}}\left(d^{-C_b+1} N_{\text{pt}}^{-1/2}\right)$ with high probability. With high probability, the following holds for all $t \in [T_1]$:

$$\left\| y^t \mathbb{1}\left[a_0 \gamma^2 + a_1 \left\langle \boldsymbol{\beta}^t, \boldsymbol{x}^t \right\rangle + a_2 \gamma \text{He}_2\left(\left\langle \boldsymbol{\beta}^t, \boldsymbol{x}^t \right\rangle\right) > 0\right] \phi\left(\boldsymbol{x}^t\right) \odot \boldsymbol{n}(\boldsymbol{Z}^t) \right\|$$

$$\leq |y^t| \left\|\boldsymbol{n}\left(\boldsymbol{Z}^t\right)\right\| \left\|\phi\left(\boldsymbol{x}^t\right)\right\|$$

$$= \tilde{\mathcal{O}}\left(d^{-C_b+2} N_{\text{pt}}^{-1/2}\right).$$

**Estimation of label-gradient correlation.** With probability at least $1 - \tilde{\mathcal{O}}\left(d^2 T_1 N_{\text{pt}}^{-\frac{1}{2}}\right)$, the following holds for all $t \in [T_1]$:

$$y^t \nabla_{\gamma} f\left(\boldsymbol{Z}^t; \boldsymbol{\gamma}(0), \boldsymbol{u}(0), \boldsymbol{v}(0), \boldsymbol{a}(0)\right)$$

$$= y^t \mathbb{1}\left[a_0 \gamma^2 + a_1 \left\langle \boldsymbol{\beta}^t, \boldsymbol{x}^t \right\rangle + a_2 \gamma \text{He}_2\left(\left\langle \boldsymbol{\beta}^t, \boldsymbol{x}^t \right\rangle\right) > 0\right] \phi(\boldsymbol{x}^t) \odot \psi\left(\boldsymbol{\beta}^t, a_0, a_1, a_2\right)$$

$$+ \tilde{\mathcal{O}}\left(d^{-C_b+2} N_{\text{pt}}^{-1/2}\right).$$

## B.2 CHARACTERIZATION OF UPDATED PARAMETER

In this step, we characterize the updated parameter by establishing concentration results.

Note that every entry of $\psi\left(\boldsymbol{\beta}^t, a_0, a_1, a_2\right)$ are $\tilde{\mathcal{O}}(d^{-C_b})$-bounded by Lemma A.5. From Lemma A.4, with high probability, we have

$$\left\|\frac{1}{T_1} \sum_{t=1}^{T_1} y^t \mathbb{1}\left[a_0 \gamma^2 + a_1 \left\langle \boldsymbol{\beta}^t, \boldsymbol{x}^t \right\rangle + a_2 \gamma \text{He}_2\left(\left\langle \boldsymbol{\beta}^t, \boldsymbol{x}^t \right\rangle\right)\right] \phi\left(\boldsymbol{x}^t\right) \odot \psi\left(\boldsymbol{\beta}^t, a_0, a_1, a_2\right) - \boldsymbol{c}\right\|$$

$$= \tilde{\mathcal{O}}\left(d^{-C_b+1} T_1^{-1/2}\right),$$

where

$$\boldsymbol{c} := \mathbb{E}\left[y^1 \mathbb{1}\left[a_0 \gamma^2 + a_1 \left\langle \boldsymbol{\beta}^1, \boldsymbol{x}^1 \right\rangle + a_2 \gamma \text{He}_2\left(\left\langle \boldsymbol{\beta}^1, \boldsymbol{x}^1 \right\rangle\right) > 0\right] \phi\left(\boldsymbol{x}^1\right) \odot \psi\left(\boldsymbol{\beta}^1, a_0, a_1, a_2\right)\right].$$

Define $B : \mathbb{R} \to \mathbb{R}$ as $B(z) = g_*(z) \mathbb{1}[a_0 \gamma^2 + a_1 z + a_2 \gamma z^2 > 0]$ and denote its Hermite expansion as $B(z) = \sum_{k=0}^{\infty} \frac{b_k}{k!} \text{He}_k(z)$. Then, we have

$$\boldsymbol{c} = \mathbb{E}_{\boldsymbol{\beta} \sim \text{Unif}(S_r)}\left[\psi\left(\boldsymbol{\beta}, a_0, a_1, a_2\right) \odot \mathbb{E}_{\boldsymbol{x} \sim \mathcal{N}(\boldsymbol{0}, \boldsymbol{I}_d)}\left[B(\boldsymbol{\beta}) \phi(\boldsymbol{x})\right]\right]$$

$$= \mathbb{E}_{\boldsymbol{\beta} \sim \text{Unif}(S_r)}\left[\psi\left(\boldsymbol{\beta}, a_0, a_1, a_2\right) \odot \psi\left(\boldsymbol{\beta}, b_0, b_1, b_2\right)\right].$$

Therefore, we conclude that

$$\frac{1}{T_1} \sum_{t=1}^{T_1} y^t \nabla_{\gamma} f\left(\boldsymbol{Z}^t; \boldsymbol{\gamma}(0), \boldsymbol{u}(0), \boldsymbol{v}(0), \boldsymbol{a}(0)\right)$$

$$= \mathbb{E}_{\boldsymbol{\beta} \sim \mathrm{Unif}(S_r)} \left[ \psi\left(\boldsymbol{\beta}, a_0, a_1, a_2\right) \odot \psi\left(\boldsymbol{\beta}, b_0, b_1, b_2\right) \right]$$
$$+ \tilde{\mathcal{O}}\left( d^{-C_b+2} N_{\mathrm{pt}}^{-1/2} \right) + \tilde{\mathcal{O}}\left( d^{-C_b+1} T_1^{-1/2} \right),$$

with probability at least $1 - \tilde{\mathcal{O}}\left( d^2 T_1 N_{\mathrm{pt}}^{-1/2} \right)$.

The remaining step is to characterize the Hermite coefficients $b_0, b_1, b_2$, and the following lemma is useful.

**Lemma B.4.** *For any non zero polynomial $P$ independent of $r$ and $d$, the following holds except for the cases $g_*$ is even function and $P$ is an odd function.:*

$$\left| \mathbb{E}_{\mathbf{z} \sim \mathcal{N}(0,1)} \left[ P(\mathbf{z}) \mathbb{1}[a_0 \gamma^2 + a_1 \mathbf{z} + a_2 \gamma \mathrm{He}_2(\mathbf{z}) > 0] \right] \right| \gtrsim \frac{1}{\mathrm{polylog}(d)}.$$

*Here, dependency of $g_*$ appears in $a_0, a_1, a_2$.*

*Proof of Lemma B.4.* Note that $e_0(g_*) = 2$. Then, from Lemma A.5, we know that $d^{C_b} a_0 = \Theta\left( (\log d)^{-2C_\rho} \right)$. In addition, for $p = 1, 2$, $d^{C_b} a_p = \Theta\left( (\log d)^{-C_\rho e_p(g_*)} \right)$ if $e_p(g_*) < \infty$ and $d^{C_b} a_p \lesssim \frac{1}{\mathrm{poly}(d)}$ otherwise.

**Case 1: $g_*$ is not an even function and $a_2 = 0$.**

In this case, $a_1 \neq 0$. We assume $a_1 > 0$, and we can also prove the case $a_1 < 0$ similarly. We have

$$\mathbb{E}_{\mathbf{z} \sim \mathcal{N}(0,1)} \left[ P(\mathbf{z}) \mathbb{1}[a_0 \gamma^2 + a_1 \mathbf{z} > 0] \right]$$
$$= \mathbb{E}_{\mathbf{z} \sim \mathcal{N}(0,1)} \left[ P(\mathbf{z}) \right] - \mathbb{E}_{\mathbf{z} \sim \mathcal{N}(0,1)} \left[ P(\mathbf{z}) \mathbb{1}[\mathbf{z} < 0] \right] - \frac{1}{\sqrt{2\pi}} \int_0^{a_0 \gamma^2 / a_1} P(z) e^{-\frac{z^2}{2}} \mathrm{d}z.$$

From our choice of $\gamma$, $a_0 \gamma^2 / a_1 = 1/\mathrm{polylog}(d)$ and we have

$$\left| \int_0^{a_0 \gamma^2 / a_1} P(z) e^{-\frac{z^2}{2}} \mathrm{d}z \right| \leq \frac{a_0 \gamma^2}{a_1} \max_{z \in [-1,1]} |P(z)| \lesssim \frac{1}{\mathrm{polylog}(d)}$$

and this provides desired conclusion for the case $\mathbb{E}_{\mathbf{z} \sim \mathcal{N}(0,1)} \left[ P(\mathbf{z}) \right] \neq \mathbb{E}_{\mathbf{z} \sim \mathcal{N}(0,1)} \left[ P(\mathbf{z}) \mathbb{1}[\mathbf{z} < 0] \right]$. For the case $\mathbb{E}_{\mathbf{z} \sim \mathcal{N}(0,1)} \left[ P(\mathbf{z}) \right] = \mathbb{E}_{\mathbf{z} \sim \mathcal{N}(0,1)} \left[ P(\mathbf{z}) \mathbb{1}[\mathbf{z} < 0] \right]$, it suffices to show that

$$\left| \int_0^{a_0 \gamma^2 / a_1} P(z) e^{-\frac{z^2}{2}} \mathrm{d}z \right| \gtrsim \frac{1}{\mathrm{polylog}(d)}.$$

Since $a_0 \gamma^2 / a_1 = 1/\mathrm{polylog}(d)$, $P(z)$ is monotone and does not change its sign in $\left[ 0, a_0 \gamma^2 / a_1 \right]$. Let $Q$ be the degree of $P$ and $q$ be the smallest degree that has non zero coefficient in $P$ and let $P(z) = \sum_{k=q}^{Q} p_k z^k$. Then, we have

$$\left| \int_0^{a_0 \gamma^2 / a_1} P(z) e^{-\frac{z^2}{2}} \mathrm{d}z \right| = \left| \int_0^{a_0 \gamma^2 / a_1} |P(z)| e^{-\frac{z^2}{2}} \mathrm{d}z \right|$$
$$\geq e^{-\frac{1}{2}} \left| \int_0^{a_0 \gamma^2 / a_1} |P(z)| \mathrm{d}z \right|$$
$$\geq e^{-\frac{1}{2}} \left| \sum_{k=q}^{Q} \frac{p_k}{k+1} \left( \frac{a_0 \gamma^2}{a_1} \right) \right|$$
$$\asymp \left( \frac{a_0 \gamma^2}{a_1} \right)^{q+1} \gtrsim \frac{1}{\mathrm{polylog}(d)}.$$

Hence, we have the desired conclusion.

**Case 2: $g_*$ is not an even function and $a_2 \neq 0$.**

In this case, $a_1, a_2 \neq 0$. We assume $a_2 > 0$ and we can prove the case $a_2 < 0$ using similar argument. Note that

$$\left| \mathbb{E}_{\mathbf{z} \sim \mathcal{N}(0,1)} \left[ P(\mathbf{z}) \mathbb{1} \left[ a_0 \gamma^2 + a_1 \mathbf{z} + a_2 \gamma \mathrm{He}_2(\mathbf{z}) > 0 \right] \right] \right|$$

$$\geq \left| \mathbb{E}_{\mathbf{z} \sim \mathcal{N}(0,1)} \left[ P(\mathbf{z}) \mathbb{1} \left[ a_0 \gamma^2 + a_1 \mathbf{z} + a_2 \gamma \mathrm{He}_2(\mathbf{z}) > 0 \wedge \mathbf{z} > -\sqrt{\log d} \right] \right] \right|$$

$$- \mathbb{E}_{\mathbf{z} \sim \mathcal{N}(0,1)} \left[ |P(\mathbf{z})| \, \mathbb{1}[z \leq -\log d] \right]$$

$$\geq \left| \mathbb{E}_{\mathbf{z} \sim \mathcal{N}(0,1)} \left[ P(\mathbf{z}) \mathbb{1} \left[ a_0 \gamma^2 + a_1 \mathbf{z} + a_2 \gamma \mathrm{He}_2(\mathbf{z}) > 0 \wedge \mathbf{z} > -\sqrt{\log d} \right] \right] \right|$$

$$- \left( \mathbb{E}_{\mathbf{z} \sim \mathcal{N}(0,1)} \left[ P(\mathbf{z})^2 \right] \right)^{\frac{1}{2}} \left( \mathbb{P}_{\mathbf{z} \sim \mathcal{N}(0,1)} \left[ z \leq -\sqrt{\log d} \right] \right)^{\frac{1}{2}}.$$

Since $\mathbb{P}_{\mathbf{z} \sim \mathcal{N}(0,1)}[z < -\log d] = o(1/\mathrm{polylog}(d))$, it suffices to show that

$$\left| \mathbb{E}_{\mathbf{z} \sim \mathcal{N}(0,1)} \left[ P(\mathbf{z}) \mathbb{1} \left[ a_0 \gamma^2 + a_1 \mathbf{z} + a_2 \gamma \mathrm{He}_2(\mathbf{z}) > 0 \wedge \mathbf{z} > -\sqrt{\log d} \right] \right] \right| \gtrsim \frac{1}{\mathrm{polylog}(d)}.$$

From our choice of $\gamma$ and Lemma A.5, we have $\theta^- := \frac{-a_1 - \sqrt{a_1^2 - 4a_2(a_0\gamma - a_2)\gamma^2}}{2a_2\gamma} < -\sqrt{\log d}$. In addition, define

$$\theta^+ := \frac{-a_1 + \sqrt{a_1^2 - 4a_2(a_0\gamma - a_2)\gamma^2}}{2a_2\gamma} = \frac{2(a_0\gamma - a_2)\gamma}{a_1 + \sqrt{a_1^2 - 4a_2(a_0\gamma - a_2)}},$$

then $|\theta^+| \lesssim 1/\mathrm{polylog}(d)$. Therefore, we have

$$\mathbb{E}_{\mathbf{z} \sim \mathcal{N}(0,1)} \left[ P(\mathbf{z}) \mathbb{1}[a_0\gamma^2 + a_1\mathbf{z} + a_2\gamma\mathrm{He}_2(\mathbf{z}) > 0 \wedge \mathbf{z} > -\sqrt{\log d}] \right]$$

$$= \mathbb{E}_{\mathbf{z} \sim \mathcal{N}(0,1)} \left[ P(\mathbf{z}) \mathbb{1} \left[ \theta^+ \leq \mathbf{z} \right] \right]$$

$$= \mathbb{E}_{\mathbf{z} \sim \mathcal{N}(0,1)} \left[ P(\mathbf{z}) \right] - \mathbb{E}_{\mathbf{z} \sim \mathcal{N}(0,1)} \left[ P(\mathbf{z}) \mathbb{1}[\mathbf{z} < 0] \right] - \frac{1}{\sqrt{2\pi}} \int_0^{\theta^+} P(z) e^{-\frac{z^2}{2}} \, \mathrm{d}z.$$

Note that

$$\left| \int_0^{\theta^+} P(z) e^{-\frac{z^2}{2}} dz \right| \leq \left| \theta^+ \right| \max_{z \in [-1,1]} |P(z)| \lesssim \frac{1}{\mathrm{polylog}(d)}$$

and this provides desired conclusion for the case $\mathbb{E}_{\mathbf{z} \sim \mathcal{N}(0,1)} \left[ P(\mathbf{z}) \right] \neq \mathbb{E}_{\mathbf{z} \sim \mathcal{N}(0,1)} \left[ P(\mathbf{z}) \mathbb{1}[\mathbf{z} < 0] \right]$. For the case $\mathbb{E}_{\mathbf{z} \sim \mathcal{N}(0,1)} \left[ P(\mathbf{z}) \right] = \mathbb{E}_{\mathbf{z} \sim \mathcal{N}(0,1)} \left[ P(\mathbf{z}) \mathbb{1}[\mathbf{z} < 0] \right]$, it suffices to show that

$$\left| \int_0^{\theta^+} P(z) e^{-\frac{z^2}{2}} dz \right| \gtrsim \frac{1}{\mathrm{polylog}(d)}.$$

Since $\theta^+ \lesssim 1/\mathrm{polylog}(d)$, $P(z)$ is monotone and does not change its sign in $[0, \theta^+]$. Let $Q$ be the degree of $P$ and $q$ be the smallest degree that has non zero coefficient in $P$ and let $P(z) = \sum_{k=q}^Q p_k z^k$. Then, we have

$$\left| \int_0^{\theta^+} P(z) e^{-\frac{z^2}{2}} \mathrm{d}z \right| = \left| \int_0^{\theta^+} |P(z)| \, e^{-\frac{z^2}{2}} \mathrm{d}z \right|$$

$$\geq e^{-\frac{1}{2}} \left| \int_0^{\theta^+} |P(z)| \, \mathrm{d}z \right|$$

$$\geq e^{-\frac{1}{2}} \left| \sum_{k=q}^Q \frac{p_k}{k+1} \left( \theta^+ \right)^k \right|$$

$$\asymp \left( \theta^+ \right)^{q+1} \gtrsim \frac{1}{\mathrm{polylog}(d)}.$$

Hence, we have desired conclusion.

**Case 3: $g_*$ is an even function.**

In this case, since $e_1(g_*) = \infty$, $|a_1| \lesssim 1/\mathrm{poly}(d)$. We assume $a_2 > 0$ and we can prove the case $a_2 < 0$ using similar arguments. Let $P_{\mathrm{even}}$ denote the even part of $P$. Then, we have

$$\mathbb{E}_{\mathbf{z} \sim \mathcal{N}(0,1)}[P(\mathbf{z})\mathbb{1}[a_0\gamma^2 + a_1\mathbf{z} + a_2\gamma \mathrm{He}_2(\mathbf{z}) > 0]]$$

$$= \mathbb{E}_{\mathbf{z} \sim \mathcal{N}(0,1)}\left[P(\mathbf{z})\mathbb{1}\left[\mathbf{z} > \theta^+ \vee \mathbf{z} < -\theta^-\right]\right]$$

$$= \mathbb{E}_{\mathbf{z} \sim \mathcal{N}(0,1)}\left[P(\mathbf{z})\mathbb{1}\left[\mathbf{z} > \sqrt{1 - a_0\gamma/a_2} \vee \mathbf{z} < -\sqrt{1 - a_0\gamma/a_2}\right]\right]$$

$$+ \frac{1}{\sqrt{2\pi}}\int_{\theta^+}^{\sqrt{1-a_0\gamma/a_2}} P(z)e^{-\frac{z^2}{2}}\,\mathrm{d}z + \frac{1}{\sqrt{2\pi}}\int_{-\sqrt{1-a_0\gamma/a_2}}^{\theta^-} P(z)e^{-\frac{z^2}{2}}\,\mathrm{d}z$$

$$= 2\mathbb{E}_{\mathbf{z} \sim \mathcal{N}(0,1)}\left[P_{\mathrm{even}}(\mathbf{z})\mathbb{1}\left[\mathbf{z} > \sqrt{1 - a_0\gamma/a_2}\right]\right]$$

$$+ \frac{1}{\sqrt{2\pi}}\int_{\theta^+}^{\sqrt{1-a_0\gamma/a_2}} P(z)e^{-\frac{z^2}{2}}\,\mathrm{d}z + \frac{1}{\sqrt{2\pi}}\int_{-\sqrt{1-a_0\gamma/a_2}}^{\theta^-} P(z)e^{-\frac{z^2}{2}}\,\mathrm{d}z$$

$$= \mathbb{E}_{\mathbf{z} \sim \mathcal{N}(0,1)}\left[P_{\mathrm{even}}(\mathbf{z})\right] - 2\mathbb{E}_{\mathbf{z} \sim \mathcal{N}(0,1)}\left[P_{\mathrm{even}}(\mathbf{z})\mathbb{1}[0 < \mathbf{z} < 1]\right]$$

$$- \frac{2}{\sqrt{2\pi}}\int_1^{\sqrt{1-a_0\gamma/a_2}} P_{\mathrm{even}}(z)e^{-\frac{z^2}{2}}\,\mathrm{d}z$$

$$+ \underbrace{\frac{1}{\sqrt{2\pi}}\int_{\theta^+}^{\sqrt{1-a_0\gamma/a_2}} P(z)e^{-\frac{z^2}{2}}\,\mathrm{d}z + \frac{1}{\sqrt{2\pi}}\int_{-\sqrt{1-a_0\gamma/a_2}}^{\theta^-} P(z)e^{-\frac{z^2}{2}}\,\mathrm{d}z}_{(*)}.$$

From our choice of $\gamma$, we have

$$\left|\int_1^{\sqrt{1-a_0\gamma/a_2}} P_{\mathrm{even}}(z)e^{-\frac{z^2}{2}}\,\mathrm{d}z\right| \leq \left|\int_1^{\sqrt{1-a_0\gamma/a_2}} |P_{\mathrm{even}}(z)|\,\mathrm{d}z\right|$$

$$\leq \left|\sqrt{1 - a_0\gamma/a_2} - 1\right| \max_{z \in [0,2]} |P_{\mathrm{even}}(z)|$$

$$\lesssim \frac{1}{\mathrm{polylog}(d)}.$$

Since $|a_1| \lesssim 1/\mathrm{poly}(d)$, we have $\left|\sqrt{1 - a_0\gamma/a_2} - \theta^+\right|, \left|-\sqrt{1 - a_0\gamma/a_2} - \theta^-\right| \lesssim 1/\mathrm{poly}(d)$ and using the same argument above, we obtain that $|(*)| \lesssim 1/\mathrm{poly}(d)$.

Hence, we obtain the conclusion if $\mathbb{E}_{\mathbf{z} \sim \mathcal{N}(0,1)}[P_{\mathrm{even}}(\mathbf{z})] \neq 2\mathbb{E}_{\mathbf{z} \sim \mathcal{N}(0,1)}[P_{\mathrm{even}}(\mathbf{z})\mathbb{1}[0 < \mathbf{z} < 1]]$. For the case $\mathbb{E}_{\mathbf{z} \sim \mathcal{N}(0,1)}[P_{\mathrm{even}}(\mathbf{z})] = 2\mathbb{E}_{\mathbf{z} \sim \mathcal{N}(0,1)}[P_{\mathrm{even}}(\mathbf{z})\mathbb{1}[0 < \mathbf{z} < 1]]$, it is enough to show that

$$\left|\int_1^{\sqrt{1-a_0\gamma/2}} P_{\mathrm{even}}(z)e^{-\frac{z^2}{2}}\,dz\right| \gtrsim \frac{1}{\mathrm{polylog}(d)}.$$

From our small choice of $\gamma$, $P_{\mathrm{even}}$ is monotone and does not change its sign in $[1, \sqrt{1 - a_0\gamma/a_2}]$ (or $[\sqrt{1 - a_0\gamma/a_2}, 1]$. Let $P_{\mathrm{even}}(z) = \sum_{k=q'}^{Q'} p_k'(z-1)^k$ with $p_{q'}', p_{Q'}' \neq 0$. Then, we have

$$\left|\int_1^{\sqrt{1-a_0\gamma/a_2}} P_{\mathrm{even}}(z)e^{-\frac{z^2}{2}}\,\mathrm{d}z\right| = \left|\int_1^{\sqrt{1-a_0\gamma/a_2}} |P_{\mathrm{even}}(z)|\,e^{-\frac{z^2}{2}}\,\mathrm{d}z\right|$$

$$\geq e^{-2}\left|\sum_{k=q'}^{Q'} \frac{p_k'}{k+1}\left(\sqrt{1 - a_0\gamma/a_2} - 1\right)^k\right|$$

$$\asymp \left|\sqrt{1 - a_0\gamma/a_2} - 1\right|^{q'} \gtrsim \frac{1}{\mathrm{polylog}(d)}.$$

Therefore, we have our desired conclusion. $\qquad\square$

By Lemma B.4 with $g_*(z), g_*(z)z, g_*(z)\mathrm{He}_2(z)$, we have $b_0, b_2 = \tilde{\Theta}(1)$ and $b_1 = \tilde{\Theta}(1)$ if $g_*$ is not an even function. We will show that $b_1 = 1/\mathrm{poly}(d)$ if $g_*$ is an even function. In this case, $a_2 \neq 0$. Without loss of generality, we assume $a_2 > 0$. Then, we have

$$2\left|\mathbb{E}_{\mathbf{z}\sim\mathcal{N}(0,1)}\left[g_*(z)z\mathbb{1}[a_0\gamma^2 + a_1\mathbf{z} + a_2\gamma\mathrm{He}_2(\mathbf{z}) > 0]\right]\right|$$

$$= \left|\mathbb{E}_{\mathbf{z}\sim\mathcal{N}(0,1)}\left[g_*(z)z\mathbb{1}[a_0\gamma^2 + a_1\mathbf{z} + a_2\gamma\mathrm{He}_2(\mathbf{z}) > 0]\right]\right.$$

$$\left. - \mathbb{E}_{\mathbf{z}\sim\mathcal{N}(0,1)}\left[g_*(z)z\mathbb{1}[a_0\gamma^2 - a_1\mathbf{z} + a_2\gamma\mathrm{He}_2(\mathbf{z}) > 0]\right]\right|$$

$$\leq \frac{1}{\sqrt{2\pi}}\int_{\theta_+^+}^{\theta_+^-}|g_*(z)z|\,\mathrm{d}z + \frac{1}{\sqrt{2\pi}}\int_{\theta_-^-}^{\theta_-^+}|g_*(z)z|\,\mathrm{d}z,$$

where $\theta_+^+, \theta_-^+, \theta_+^-$, and $\theta_-^-$ are defined as follows:

$$\theta_+^+ := \frac{|a_1|}{2a_2\gamma} + \sqrt{\left(\frac{a_1}{2a_2\gamma}\right)^2 + 1 - \frac{a_1\gamma}{a_2}}, \quad \theta_-^+ := -\frac{|a_1|}{2a_2\gamma} + \sqrt{\left(\frac{a_1}{2a_2\gamma}\right)^2 + 1 - \frac{a_1\gamma}{a_2}}$$

$$\theta_+^- := \frac{|a_1|}{2a_2\gamma} - \sqrt{\left(\frac{a_1}{2a_2\gamma}\right)^2 + 1 - \frac{a_1\gamma}{a_2}}, \quad \theta_-^- := -\frac{|a_1|}{2a_2\gamma} - \sqrt{\left(\frac{a_1}{2a_2\gamma}\right)^2 + 1 - \frac{a_1\gamma}{a_2}}.$$

From Lemma A.5 and our choice of $\gamma$, $[\theta_-^+, \theta_+^+] \subset [0, 2]$. Hence, we have

$$\int_{\theta_-^+}^{\theta_+^+}|g_*(z)z|\,\mathrm{d}z \leq \left(\theta_+^+ - \theta_-^+\right)\max_{z\in[0,2]}|g_*(z)z| = \frac{a_1}{a_2\gamma}\max_{z\in[0,2]}|g_*(z)z| = \frac{1}{\mathrm{poly}(d)}.$$

Applying a similar argument, we have the same bound for the second term, and we conclude $b_1 = 1/\mathrm{poly}(d)$.

**Updated Parameter $\gamma^*$.** With probability at least $1 - \tilde{\mathcal{O}}\left(d^2 T_1 N_{\mathrm{pt}}^{-1/2}\right)$, the updated parameter $\gamma^*$ is given by:

$$\gamma^* = 2\eta\mathbb{E}_{\boldsymbol{\beta}\sim\mathrm{Unif}(S_r)}\left[\psi(\boldsymbol{\beta}, a_0, a_1, a_2) \odot \psi(\boldsymbol{\beta}, b_0, b_1, b_2)\right]$$

$$+ \eta\tilde{\mathcal{O}}\left(\max\left\{d^{-2C_b+4}, d^{-C_b+2}N_{\mathrm{pt}}^{-1/2}, d^{-C_b+1}T_1^{-1/2}\right\}\right), \tag{4}$$

with $a_0, b_0, b_2 = \tilde{\Theta}(1)$ and $a_{\mathrm{ge}(g_*)} = \tilde{\Theta}(1), a_{3-\mathrm{ge}(g_*)} = \tilde{\mathcal{O}}(1)$. Furthermore, $b_1 = \tilde{\Theta}(1)$ if $\mathrm{ge}(g_*) = 1$, and $b_1 \lesssim 1/\mathrm{poly}(d)$ otherwise.

### B.3 OUTPUT OF UPDATED MAMBA LAYER

Lastly, we characterize the output of the Mamba layer with the updated parameter $\gamma^*$, which serves as the input to the MLP layer. This characterization is given in the following proposition, which is a formal statement of Proposition 4.1.

**Proposition B.5.** *Let* $(\boldsymbol{x}_1, y_1, \ldots, \boldsymbol{x}_N, y_N, \boldsymbol{x})$ *be a prompt with context length* $N \leq N^*$ *and feature vector* $\boldsymbol{\beta} \in \mathbb{R}^d$ *and its embedding* $\boldsymbol{Z} \in \mathbb{R}^{(\tilde{d}+1)\times(N+1)}$. *If* $N = \tilde{\Omega}\left(r^{3\mathrm{ge}(g_*)}\right)$, *updated parameter* $\gamma^*$ *satisfies (4), and* $\eta = \Theta(d^{2C_b}(\log d)^{-C_\eta})$ *with some large enough constant* $C_\eta > 0$, *then the following holds with high probability:*

$$N^{-1}\mathsf{Mamba}\left(\boldsymbol{Z}; \gamma^*\right)[\tilde{d}+1, N+1] = P_1 + P_2\left(\frac{\langle\boldsymbol{\beta}, \boldsymbol{x}\rangle}{r}\right)^{\mathrm{ge}(g_*)} + o\left(P_2 r^{-3\mathrm{ge}(g_*)/2}(\log d)^{-2\deg(g_*)+2}\right),$$

*where* $P_1$ *and* $P_2$ *are independent of data and satisfies* $P_1 = o(1)$ *and* $P_2 = \Theta((\log d)^{-C_{P_2}})$ *with some constant* $C_{P_2} > 0$.

*Proof of Proposition B.5.* Recall that $\boldsymbol{c} := \mathbb{E}_{\boldsymbol{\beta}\sim\mathrm{Unif}(S_r)}\left[\psi(\boldsymbol{\beta}, a_0, a_1, a_2) \odot \psi(\boldsymbol{\beta}, b_0, b_1, b_2)\right]$. From (4), we have

$$(2\eta)^{-1}N^{-1}\mathsf{Mamba}\left(\boldsymbol{Z}; \gamma^*\right)[\tilde{d}+1, N+1]$$

$$= N^{-1}\sum_{j=1}^{N} G_{j,N+1}\left(\boldsymbol{Z}\right)y_j\phi\left(\boldsymbol{x}_j\right)^\top\left(\boldsymbol{c} \odot \phi\left(\boldsymbol{x}\right)\right)$$

$$+ \tilde{\mathcal{O}} \left( \max \left\{ d^{-3C_b+4}, d^{-2C_b+4} N_{\mathrm{pt}}^{-1/2}, d^{-2C_b+3} T_1^{-1/2} \right\} \right)$$

$$= N^{-1} \sum_{j=1}^{N} y_j \sigma(y_j/\rho + b) \phi\left( \boldsymbol{x}_j \right)^{\top} \left( \boldsymbol{c} \odot \phi\left( \boldsymbol{x} \right) \right)$$

$$- N^{-1} \sum_{j=1}^{N} \left[ y_j \sigma(y_j/\rho + b) \left( 1 - \left( 1 - \sigma(b) \right) \prod_{i=j+1}^{N} \left( 1 - \sigma(y_j^t/\rho + b) \right) \right) \phi\left( \boldsymbol{x}_j^t \right)^{\top} \left( \boldsymbol{c} \odot \phi(\boldsymbol{x}) \right) \right]$$

$$+ \tilde{\mathcal{O}} \left( \max \left\{ d^{-3C_b+4}, d^{-2C_b+4} N_{\mathrm{pt}}^{-1/2}, d^{-2C_b+3} T_1^{-1/2} \right\} \right).$$

Note that for each $j \in [N]$, with high probability, $y_j = \rho \bar{g}_*(\boldsymbol{\beta}, \boldsymbol{x}_j) + \zeta_i$ where $\zeta_i \sim \mathrm{Unif}(\{-\tau, \tau\})$. It implies that

$$\left| y_j \sigma(y_j/\rho + b) \left( 1 - \left( 1 - \sigma(b) \right) \prod_{i=j+1}^{N} \left( 1 - \sigma(y_j^t/\rho + b) \right) \right) \phi\left( \boldsymbol{x}_j^t \right)^{\top} \left( \boldsymbol{c} \odot \phi(\boldsymbol{x}) \right) \right|$$

$$\leq N^{-1} \sum_{j=1}^{N} \left[ |y_j \sigma(y_j/\rho + b)| \left( 1 - \left( 1 - \sigma(2b) \right)^{N^*} \right) \| \phi\left( \boldsymbol{x}_j \right) \| \right] \|\boldsymbol{c}\| \|\phi(\boldsymbol{x})\|$$

$$= \tilde{\mathcal{O}} \left( d^{(-3C_b + C^* + 2)} \right).$$

In addition, with high probability, we have

$$\phi\left( \boldsymbol{x}_j \right)^{\top} \left( \boldsymbol{c} \odot \phi(\boldsymbol{x}) \right)$$

$$= \mathbb{E}_{\boldsymbol{\beta} \sim \mathrm{Unif}(S_r)} \left[ \langle \psi(\boldsymbol{\beta}, a_0, a_1, a_2) \odot \phi(\boldsymbol{x}_j), \psi(\boldsymbol{\beta}, b_0, b_1, b_2) \odot \phi(\boldsymbol{x}) \rangle \right]$$

$$= a_0 b_0 + a_1 b_1 \mathbb{E}_{\boldsymbol{\beta} \sim \mathrm{Unif}(S_r)} [\langle \boldsymbol{\beta}, \boldsymbol{x}_i \rangle \langle \boldsymbol{\beta}, \boldsymbol{x} \rangle]$$

$$+ \frac{a_2 b_2}{2} \mathbb{E}_{\boldsymbol{\beta} \sim \mathrm{Unif}(S_r)} \left[ (\mathrm{He}_2(\langle \boldsymbol{\beta}, \boldsymbol{x} \rangle) - 1)(\mathrm{He}_2(\langle \boldsymbol{\beta}, \boldsymbol{x}_j \rangle) - 1) \right].$$

In addition, combining with Lemma A.3, we have

$$\phi\left( \boldsymbol{x}_j \right)^{\top} \left( \boldsymbol{c} \odot \phi\left( \boldsymbol{x} \right) \right) = \tilde{\mathcal{O}} \left( d^{-C_b} \right),$$

with high probability. Therefore, from our choice of $\eta = \Theta \left( d^{2C_b} (\log d)^{-C_\eta} \right)$, with high probability, we have

$$\left| \eta y_j \sigma(y_j/\rho + b) \phi(\boldsymbol{x}_j)^{\top} \left( \boldsymbol{c} \odot \phi(\boldsymbol{x}) \right) \right| \leq 1.$$

Therefore, with high probability, we have

$$N^{-1} \sum_{j=1}^{N} \eta y_j \sigma(y_j/\rho + b) \phi(\boldsymbol{x}_j)^{\top} \left( \boldsymbol{c} \odot \phi(\boldsymbol{x}) \right) = N^{-1} \sum_{j=1}^{N} \bar{\mathbf{z}}_j,$$

where

$$\mathbf{z}_j := \eta y_j \sigma(y_j/\rho + b) \phi(\boldsymbol{x}_j)^{\top} \left( \boldsymbol{c} \odot \phi(\boldsymbol{x}) \right), \quad \bar{\mathbf{z}}_j := \mathbf{z}_j \mathbb{1}[|\mathbf{z}_j| \leq 1].$$

By Höeffding's inequality, with high probability, we have

$$N^{-1} \sum_{j=1}^{N} \bar{\mathbf{z}}_j = \mathbb{E}_{\boldsymbol{x}_1, y_1}[\bar{\mathbf{z}}_1] + \tilde{\mathcal{O}} \left( N^{-1/2} \right)$$

$$= \mathbb{E}_{\mathbf{x}_1, y_1}[\mathbf{z}_1] + \mathbb{E}_{\mathbf{x}_1, y_1} \left[ \mathbf{z}_1 \mathbb{1} \left[ |\mathbf{z}_1| > r^2 \right] \right] + \tilde{\mathcal{O}} \left( N^{-1/2} \right)$$

$$= \mathbb{E}_{\mathbf{x}_1, y_1}[\mathbf{z}_1] + \tilde{\mathcal{O}} \left( N^{-1/2} \right),$$

where the last inequality holds since

$$\left| \mathbb{E}_{\mathbf{x}_1, y_1} \left[ \mathbf{z}_1 \mathbb{1} \left[ |\mathbf{z}_1| > r^2 \right] \right] \right| \leq \mathbb{E}_{\mathbf{x}_1, y_1} \left[ \mathbf{z}_1^2 \right]^{\frac{1}{2}} \mathbb{P} \left[ |\mathbf{z}_1| > r^2 \right]^{\frac{1}{2}} = \frac{1}{\mathrm{poly}(d)}.$$

Therefore, with high probability, we have

$$N^{-1}\text{Mamba}\left(\boldsymbol{Z};\boldsymbol{\gamma}^*\right)[\tilde{d}+1,N+1]$$
$$= 2\eta\mathbb{E}\left[y_1\sigma(y_1/\rho+b)\phi(\boldsymbol{x}_1)^\top\left(\boldsymbol{c}\odot\phi(\boldsymbol{x})\right)\right]$$
$$+ \tilde{\mathcal{O}}\left(N^{-1/2}\right) + \tilde{\mathcal{O}}\left(\max\left\{d^{-C_b+6},d^4N_{\text{pt}}^{-1/2},d^3T_1^{-1/2}\right\}\right).$$

Lastly, the expectation can be calculated as

$$\psi\left(\boldsymbol{\beta},a_0,a_1,a_2\right)^\top\left(\boldsymbol{c}\odot\phi\left(\boldsymbol{x}\right)\right)$$
$$= \mathbb{E}_{\boldsymbol{\beta}'\sim\text{Unif}(S_r)}\left[\left\langle\psi(\boldsymbol{\beta},a_0,a_1,a_2)\odot\psi(\boldsymbol{\beta}',a_0,a_1,a_2),\psi(\boldsymbol{\beta}',b_0,b_1,b_2)\odot\phi(\boldsymbol{x})\right\rangle\right]$$
$$= a_0^2b_0 + a_1^2b_1\mathbb{E}_{\boldsymbol{\beta}'\sim\text{Unif}(S_r)}\left[\sum_{i=1}^d\boldsymbol{\beta}[i]\boldsymbol{x}[i]\boldsymbol{\beta}'[i]^2\right]$$
$$+ \frac{a_2^2b_2}{4}\left(\left(\sum_{i=1}^d\boldsymbol{\beta}[i]\boldsymbol{x}[i]\boldsymbol{\beta}'[i]^2\right)^2 - \sum_{i=1}^d\boldsymbol{\beta}[i]^2\boldsymbol{\beta}'[i]^2\right)$$
$$= \left(a_0^2b_0 - \frac{a_2^2b_2}{4}\right) + a_1^2b_1\left(\frac{\langle\boldsymbol{\beta},\boldsymbol{x}\rangle}{r}\right) + \frac{a_2^2b_2}{4}\left(\frac{\langle\boldsymbol{\beta},\boldsymbol{x}\rangle}{r}\right)^2.$$

Hence, we have

$$N^{-1}\text{Mamba}\left(\boldsymbol{Z};\boldsymbol{\gamma}^*\right)[\tilde{d}+1,N+1]$$
$$= 2\eta\left(\left(a_0^2b_0 - \frac{a_2^2b_2}{4}\right) + a_1^2b_1\left(\frac{\langle\boldsymbol{\beta},\boldsymbol{x}\rangle}{r}\right) + \frac{a_2^2b_2}{4}\left(\frac{\langle\boldsymbol{\beta},\boldsymbol{x}\rangle}{r}\right)^2\right)$$
$$+ \tilde{\mathcal{O}}\left(N^{-\frac{1}{2}}\right) + \tilde{\mathcal{O}}\left(\max\left\{d^{-C_b+6},d^4N_{\text{pt}}^{-1/2},d^3T_1^{-1/2}\right\}\right).$$

Our conclusion is reached by defining $P_1 = 2\eta(a_0^2b_0 - a_2^2b_2/4)$ and

$$P_2 = \begin{cases} 2\eta a_1^2b_1 & \text{if } \text{ge}(g_*) = 1 \\ \eta a_2^2b_2/2 & \text{if } \text{ge}(g_*) = 2 \end{cases}.$$

$\square$

## C  OPTIMIZING MLP LAYER

In this section, we analyze the second stage of pretraining, which focuses on the MLP layer.

### C.1  CONSTRUCTION OF APPROXIMATING MLP LAYER

First, we construct the infinite-width MLP layer approximating the link function $g_*$.

**Lemma C.1.** *For given $\boldsymbol{\beta}\in\mathbb{R}^d$ with $\|\boldsymbol{\beta}\| = 1$, suppose there exists a function $h:\mathbb{R}\to\mathbb{R}$ such that*

$$h(\boldsymbol{x}) = P_1 + P_2\left(\frac{\langle\boldsymbol{\beta},\boldsymbol{x}\rangle}{r}\right)^{\text{ge}(g_*)} + n(\boldsymbol{x}),$$

*where $P_1 = o(1), P_2 = \Theta\left((\log d)^{-C_{P_2}}\right)$, and $|n(\boldsymbol{x})| = o\left(P_2r^{-3\text{ge}(g_*)/2}(\log d)^{-2\deg(g_*)+2}\right)$ with high probability over $\boldsymbol{x}\sim\mathcal{N}(\boldsymbol{0},\boldsymbol{I}_d)$. Then, there exists a function $\pi(\cdot,\cdot):\mathbb{R}^2\to\mathbb{R}$ such that*

$$\left|\mathbb{E}_{v\sim\text{Unif}(\{\pm1\}),a\sim\text{Unif}([-1,1])}[\phi(v,a)\text{ReLU}(vh(\boldsymbol{x})+a) - g_*(\langle\boldsymbol{\beta},\boldsymbol{x}\rangle)]\right| = o(1),$$

*with high probability over $\boldsymbol{x}\sim\mathcal{N}(\boldsymbol{0},\boldsymbol{I}_d)$. In addition, $\sup_{v,a}|\pi(v,a)| = \tilde{\mathcal{O}}(r^{2\text{ge}(g_*)})$.*

*Proof of Lemma C.1.* Since $\text{ge}(g_*) = 2$ implies $g_*$ is even function, there exists a polynomial $\tilde{g}_*$ such that $g_*(z) = \tilde{g}_*\left(z^{\text{ge}(g_*)}\right)$. Let $\tilde{g}_*(z) = \sum_{k=0}^{\deg(\tilde{g}_*)}s_kz^k$. For any $k\in\mathbb{N}_0$, from Lemma 17 in Damian et al. (2022), there exists $\pi_k(\cdot,\cdot):\mathbb{R}^2\to\mathbb{R}$ such that for any $|z|\leq 1$

$$\mathbb{E}_{v\sim\text{Unif}(\{\pm1\}),a\sim\text{Unif}([-1,1])}\left[\pi_k'(v,a)\text{ReLU}(vz+a)\right] = z^k \text{ and } \sup_{v,a}|\pi_k'(v,a)| = \mathcal{O}(1).$$

Let us define $\pi'(\cdot, \cdot) : \mathbb{R}^2 \to \mathbb{R}$ as

$$\pi'(v, a) = \sum_{k=0}^{\deg(\tilde{g}_*)} s_k \frac{\pi'_k\left(v, ap^{-1}(\log d)^{-2}\right)}{p(\log d)^2} (\log d)^{2k},$$

where $p := P_2 r^{-\mathrm{ge}(g_*)}$. Note that $\sup_{v,a} |\pi'(v, a)| = \mathcal{O}\left(p^{-1}(\log d)^{2\deg(g_*)-2}\right)$ and if $|z| \le (\log d)^2$, then we have

$$\mathbb{E}_{\substack{v \sim \mathrm{Unif}(\{\pm 1\}) \\ a \sim \mathrm{Unif}([-p(\log d)^2, p(\log d)^2])}} [\pi'(v, a)\mathrm{ReLU}(v(pz) + a)]$$

$$= \sum_{k=0}^{\deg(\tilde{g}_*)} s_k \mathbb{E}_{\substack{v \sim \mathrm{Unif}(\{\pm 1\}) \\ a \sim \mathrm{Unif}([-p(\log d)^2, p(\log d)^2])}} \left[ \frac{\pi'_k(v, ap^{-1}(\log d)^{-2})}{p(\log d)^2}(\log d)^{2k}\mathrm{ReLU}(v(pz) + a) \right]$$

$$= \sum_{k=0}^{\deg(\tilde{g}_*)} s_k (\log d)^{2k} \mathbb{E}_{v \sim \mathrm{Unif}(\{\pm 1\}), a \sim \mathrm{Unif}([-1,1])} [\pi'_k(v, a)\mathrm{ReLU}(vz(\log d)^{-2} + a)]$$

$$= \sum_{k=0}^{\deg(\tilde{g}_*)} s_k z^k = \tilde{g}_*(z).$$

Lastly, we define $\pi(\cdot, \cdot) : \mathbb{R}^2 \to \mathbb{R}$ as

$$\pi(v, a) := \frac{\mathbb{1}[v = -1 \wedge a \in [P_1 - p(\log d)^2, P_1 + p(\log d)^2]]\pi'(-1, b - P_1)}{2p(\log d)^2}$$

$$+ \frac{\mathbb{1}[v = 1 \wedge a \in [-P_1 - p(\log d)^2, -P_1 + p(\log d)^2]]\pi'(1, b + P_1)}{2p(\log d)^2}, \quad (5)$$

then we have $\sup_{v,a} = \tilde{\mathcal{O}}(r^{2\mathrm{ge}(g_*)})$ With high probability, $|\langle \boldsymbol{\beta}, \boldsymbol{x} \rangle| \le (\log d)^2$ and thus we have

$$2\mathbb{E}_{v \sim \mathrm{Unif}(\{\pm 1\}), a \sim \mathrm{Unif}([-1,1])}[\pi(v, a)\mathrm{ReLU}(vh(\boldsymbol{x}) + b)]$$

$$= \mathbb{E}_{a \sim \mathrm{Unif}([P_1 - p(\log d)^2, P_1 + p(\log d)^2])} \left[ \pi'(-1, b - P_1)\mathrm{ReLU}(-P_1 - p(\langle \boldsymbol{\beta}, \boldsymbol{x} \rangle)^{\mathrm{ge}(g_*)}) - n(\boldsymbol{x}) + a \right]$$

$$+ \mathbb{E}_{a \sim \mathrm{Unif}([P_1 - p(\log d)^2, P_1 + p(\log d)^2])} \left[ \pi'(1, b - P_1)\mathrm{ReLU}(P_1 + p(\langle \boldsymbol{\beta}, \boldsymbol{x} \rangle)^{\mathrm{ge}(g_*)} + n(\boldsymbol{x}) + a) \right]$$

$$= 2\mathbb{E}_{v \sim \mathrm{Unif}(\{\pm 1\}), a \sim \mathrm{Unif}([-p(\log d)^2, p(\log d)^2])} \left[ \pi'(v, a)\mathrm{ReLU}\left(vp\langle \boldsymbol{\beta}, \boldsymbol{x} \rangle^{\mathrm{ge}(g_*)} + a\right) \right] + o(1)$$

$$= 2g_*(\langle \boldsymbol{\beta}, \boldsymbol{x} \rangle) + o(1).$$

Here, the second equality holds from the fact that $\sup_{v,a} |\pi'(v, a)| = \mathcal{O}\left(p^{-1}(\log d)^{2\deg(g_*)-2}\right)$ and $|n(\boldsymbol{x})| = o\left(p(\log d)^{-2\deg(g_*)+2}\right)$.

Therefore, we have the desired conclusion. □

Next, we prove that we can approximate the link function with a finite-width MLP.

**Lemma C.2.** *Let $\boldsymbol{v} \sim \mathrm{Unif}(\{\pm 1\}^m)$ and $\boldsymbol{a} \sim \mathrm{Unif}([-1, 1]^m)$. Under the same condition of Lemma C.1, there exists $\boldsymbol{u}' \in \mathbb{R}^m$ such that*

$$\left| \sum_{j=1}^m \boldsymbol{u}'[j]\mathrm{ReLU}(\boldsymbol{v}[j]h(\boldsymbol{x}) + \boldsymbol{a}[j]) - g_*(\langle \boldsymbol{\beta}, \boldsymbol{x} \rangle) \right| = o(1)$$

*with high probability over $\boldsymbol{x} \sim \mathcal{N}(\boldsymbol{0}, \boldsymbol{I}_d)$. Furthermore, $\|\boldsymbol{u}'\|^2 = \tilde{\mathcal{O}}\left(r^{3\mathrm{ge}(g_*)}m^{-1}\right)$ holds with high probability.*

*Proof of Lemma C.2.* We choose $\boldsymbol{u}'$ as $\boldsymbol{u}'[j] = \pi(\boldsymbol{v}[j], \boldsymbol{a}[j])/m$ where $\pi(\cdot, \cdot) : \mathbb{R}^2 \to \mathbb{R}$ is obtained from Lemma C.1. We will show that this choice satisfies the desired conclusions. Since $|h(\boldsymbol{x})| \le 1$ with high probability and $\sup_{v,a} |\pi(v, a)| = \tilde{\mathcal{O}}(r^{2\mathrm{ge}(g_*)})$, we can apply Höeffding's inequality:

$$\sum_{j=1}^m \boldsymbol{u}'[j]\mathrm{ReLU}(\boldsymbol{v}[j]h(\boldsymbol{x}) + \boldsymbol{a}[j])$$

$$= \frac{1}{m} \sum_{j=1}^{m} \pi(\boldsymbol{v}[j], \boldsymbol{a}[j]) \text{ReLU}(\boldsymbol{v}[j]h(\boldsymbol{x}) + \boldsymbol{a}[j])$$

$$= \mathbb{E}_{v \sim \text{Unif}(\{\pm 1\}), a \sim \text{Unif}([-1,1])}[\pi(v,a)\text{ReLU}(vh(\boldsymbol{x}) + a)] + \tilde{\mathcal{O}}(r^{2\text{ge}(g_*)}m^{-1/2})$$

$$= g_*(\langle \boldsymbol{\beta}, \boldsymbol{x} \rangle) + \tilde{\mathcal{O}}(r^{2\text{ge}(g_*)}m^{-1/2}) + o(1).$$

In addition, by applying Höeffding's inequality, the following holds with high probability:

$$\|\boldsymbol{u}'\|^2 = m^{-2} \sum_{j=1}^{m} \pi(\boldsymbol{v}[j], \boldsymbol{b}[i])^2$$

$$= m^{-1}\mathbb{E}_{v \sim \text{Unif}(\{\pm 1\}), a \sim \text{Unif}([-1,1])}[\pi(v,a)^2] + \tilde{\mathcal{O}}(r^{4\text{ge}(g_*)}m^{-3/2}).$$

From (5), $\pi(v, a)$ is nonzero with probability $\tilde{\mathcal{O}}(r^{-\text{ge}(g_*)})$. Combining with $\sup_{v,a} |\pi(v,a)| = \tilde{\mathcal{O}}(r^{2\text{ge}(g_*)})$, we have desired conclusion. $\square$

## C.2 CHARACTERIZATION OF ESTIMATION ERROR ON THE TRAINING SET

The following lemma characterizes estimation on the training set after pretraining.

**Lemma C.3.** *There exists $\lambda_2 > 0$ such that the following holds with probability at least $0.999$:*

$$\frac{1}{T_2} \sum_{t=T_1+1}^{T_1+T_2} \left| y^t - f(\boldsymbol{Z}^t, \boldsymbol{\gamma}^*, \boldsymbol{u}^*, \boldsymbol{v}^*, \boldsymbol{a}^*) \right| = \tau + o(1) \text{ and } \|\boldsymbol{u}^*\| = \tilde{\mathcal{O}}\left(r^{3\text{ge}(g^*)/2}m^{-\frac{1}{2}}\right).$$

*Proof of Lemma C.3.* From Proposition B.5, the condition in Lemma C.1 is satisfied with probability at least $0.999$. Under this, let $\boldsymbol{u}'$ be the output layer parameter obtained from Lemma C.2. From the equivalence between $\ell_2$-regularization and norm-constrained optimization, there exists $\lambda_2 > 0$ such that optimized parameter $\boldsymbol{u}^*$ satisfies $\|\boldsymbol{u}^*\| \leq \|\boldsymbol{u}'\| = \tilde{\mathcal{O}}(r^{3\text{ge}(g_*)/2}m^{-1/2})$ and

$$\left( \frac{1}{T_2} \sum_{t=T_1+1}^{T_1+T_2} \left| y^t - f(\boldsymbol{Z}^t, \boldsymbol{\gamma}^*, \boldsymbol{u}^*, \boldsymbol{v}^*, \boldsymbol{a}^*) \right| \right)^2 \leq \frac{1}{T_2} \sum_{t=T_1+1}^{T_1+T_2} \left( y^t - f(\boldsymbol{Z}^t, \boldsymbol{\gamma}^*, \boldsymbol{u}^*, \boldsymbol{v}^*, \boldsymbol{a}^*) \right)^2$$

$$\leq \frac{1}{T_2} \sum_{t=T_1+1}^{T_1+T_2} \left( y^t - f(\boldsymbol{Z}^t, \boldsymbol{\gamma}^*, \boldsymbol{u}', \boldsymbol{v}^*, \boldsymbol{a}^*) \right)^2$$

$$\leq (\tau + o(1))^2.$$

$\square$

# D TEST ERROR ANALYSIS

In this section, we analyze the test-time estimation error:

$$\mathcal{R}_{N_{\text{test}}}(\boldsymbol{\gamma}^*, \boldsymbol{u}^*, \boldsymbol{v}^*, \boldsymbol{a}^*) = \mathbb{E}_{(\boldsymbol{Z},y) \sim \mathcal{D}(N_{\text{test}})}[|f(\boldsymbol{Z}, \boldsymbol{\gamma}^*, \boldsymbol{u}^*, \boldsymbol{v}^*, \boldsymbol{a}^*) - y|].$$

## D.1 TEST ERROR FOR PROMPTS WITH PRETRAINING CONTEXT LENGTH

We first prove our conclusion for the case $N_{\text{test}} = N_{\text{pt}}$ by establishing a generalization bound using Rademacher complexity.

We define a family of functions $\mathcal{F}_U$ on inputs with context length $N_{\text{test}} = N_{\text{pt}}$ as follows:

$$\mathcal{F}_U := \left\{ (\boldsymbol{Z}, y) \mapsto \sum_{j=1}^{m} \boldsymbol{u}[j]\text{ReLU}\left(\boldsymbol{v}^*[j]N_{\text{pt}}^{-1}\text{Mamba}(\boldsymbol{Z}; \boldsymbol{\gamma}^*) + \boldsymbol{a}^*[j]\right) \middle| \|\boldsymbol{u}\| \leq U \right\}.$$

In addition, the Rademacher complexity of $\mathcal{F}_U$ for sample size $T_2$ is defined as

$$\text{Rad}_{T_2}(\mathcal{F}_U) = \mathbb{E}_{\substack{(\boldsymbol{Z}^t, y^t) \sim \mathcal{D}(N_{\text{pt}}) \\ \boldsymbol{\epsilon} \sim \text{Unif}(\{\pm 1\}^{T_2})}} \left[ \sup_{\tilde{f} \in \mathcal{F}_U} \frac{1}{T_2} \sum_{t=1}^{T_2} \boldsymbol{\epsilon}[t]\tilde{f}(\boldsymbol{Z}^t, y^t) \right].$$

In the following lemma, we characterize the Rademacher complexity of $\mathcal{F}_U$.

**Lemma D.1.** *It holds that*

$$\text{Rad}_{T_2}(\mathcal{F}_U) = \tilde{\mathcal{O}}\left(U m^{1/2} T_2^{-1/2}\right).$$

*Proof of Lemma D.1.* By sequentially applying Cauchy-Schwarz inequality and Jensen's inequality, we have

$$\text{Rad}_{T_2}(\mathcal{F}_U)$$

$$= \mathbb{E}_{\substack{(\boldsymbol{Z}^t, y^t) \sim \mathcal{D}(N_{\text{pt}}) \\ \boldsymbol{\epsilon} \sim \text{Unif}(\{\pm 1\}^{T_2})}} \left[ \sup_{\|\boldsymbol{u}\| \leq U} \sum_{j=1}^{m} \boldsymbol{u}[j] \left( \frac{1}{T_2} \sum_{t=1}^{T_2} \boldsymbol{\epsilon}[t] \text{ReLU}\left( \boldsymbol{v}^*[j] N_{\text{pt}}^{-1} \text{Mamba}(\boldsymbol{Z}^t, \boldsymbol{\gamma}^*) + \boldsymbol{a}^*[j] \right) \right) \right]$$

$$\leq A \mathbb{E}_{\substack{(\boldsymbol{Z}^t, y^t) \sim \mathcal{D}(N_{\text{pt}}) \\ \boldsymbol{\epsilon} \sim \text{Unif}(\{\pm 1\}^{T_2})}} \left[ \left( \sum_{j=1}^{m} \left( \frac{1}{T_2} \sum_{t=1}^{T_2} \boldsymbol{\epsilon}[t] \text{ReLU}\left( \boldsymbol{v}^*[j] N_{\text{pt}}^{-1} \text{Mamba}(\boldsymbol{Z}^t, \boldsymbol{\gamma}^*) + \boldsymbol{a}^*[j] \right) \right)^2 \right)^{\frac{1}{2}} \right]$$

$$\leq A \left( \mathbb{E}_{\substack{(\boldsymbol{Z}^t, y) \sim \mathcal{D}(N_{\text{pt}}) \\ \boldsymbol{\epsilon} \sim \text{Unif}(\{\pm 1\}^{T_2})}} \left[ \sum_{j=1}^{m} \left( \frac{1}{T_2} \sum_{t=1}^{T_2} \boldsymbol{\epsilon}[t] \text{ReLU}\left( \boldsymbol{v}^*[j] N_{\text{pt}}^{-1} \text{Mamba}(\boldsymbol{Z}^t, \boldsymbol{\gamma}^*) + \boldsymbol{a}^*[j] \right) \right)^2 \right] \right)^{\frac{1}{2}}$$

$$= A \left( \mathbb{E}_{(\boldsymbol{Z}^t, y^t) \sim \mathcal{D}(N_{\text{pt}})} \left[ \frac{1}{T^2} \sum_{j=1}^{m} \sum_{t=1}^{T_2} \left( \text{ReLU}\left( \boldsymbol{v}^*[j] N_{\text{pt}}^{-1} \text{Mamba}(\boldsymbol{Z}^t, \boldsymbol{\gamma}^*) + \boldsymbol{a}^*[j] \right) \right)^2 \right] \right)^{\frac{1}{2}}.$$

In addition, we have

$$\mathbb{E}_{(\boldsymbol{Z}^t, y^t) \sim \mathcal{D}(N_{\text{pt}})} \left[ \sum_{j=1}^{m} \sum_{t=1}^{T_2} \left( \text{ReLU}\left( \boldsymbol{v}^*[j] N_{\text{pt}}^{-1} \text{Mamba}(\boldsymbol{Z}^t, \boldsymbol{\gamma}^*) + \boldsymbol{a}^*[j] \right) \right)^2 \right]$$

$$\leq \mathbb{E}_{(\boldsymbol{Z}^t, y^t) \sim \mathcal{D}(N_{\text{pt}})} \left[ \sum_{j=1}^{m} \sum_{t=1}^{T_2} \left( \boldsymbol{v}^*[j] N_{\text{pt}}^{-1} \text{Mamba}(\boldsymbol{Z}^t, \boldsymbol{\gamma}^*) + \boldsymbol{a}^*[j] \right)^2 \right]$$

$$\leq 2 \mathbb{E}_{(\boldsymbol{Z}^t, y^t) \sim \mathcal{D}(N_{\text{pt}})} \left[ \sum_{j=1}^{m} \sum_{t=1}^{T_2} \left( \boldsymbol{v}^*[j]^2 N_{\text{pt}}^{-2} \text{Mamba}(\boldsymbol{Z}^t, \boldsymbol{\gamma}^*)^2 + \boldsymbol{a}^*[j]^2 \right) \right]$$

$$\leq 2 \left( m T_2 + m T_2 \mathbb{E}_{(\boldsymbol{Z}, y) \sim \mathcal{D}(N_{\text{pt}})} \left[ N_{\text{pt}}^{-2} \text{Mamba}(\boldsymbol{Z}, \boldsymbol{\gamma}^*)^2 \right] \right).$$

Let $(\boldsymbol{Z}, y) \sim \mathcal{D}(N_{\text{pt}})$ and $\boldsymbol{\beta}$, $\boldsymbol{x}$ be their feature vector and query data, respectively. From Lemma A.1, with high probability over $(\boldsymbol{Z}, y) \sim \mathcal{D}(N_{\text{pt}})$, we have

$$N_{\text{pt}}^{-1} \text{Mamba}(\boldsymbol{Z}, \boldsymbol{\gamma}^*) = P_1 + P_2 \left( \frac{\langle \boldsymbol{\beta}, \boldsymbol{x} \rangle}{r} \right)^{\text{ge}(g_*)} + o\left( P_2 r^{-\text{ge}(g_*)} (\log d)^{-2 \deg(g_*)} \right) = \tilde{\mathcal{O}}(1).$$

It implies $\mathbb{E}_{(\boldsymbol{Z}, y) \sim \mathcal{D}(N_{\text{pt}})} \left[ N_{\text{pt}}^{-2} \text{Mamba}(\boldsymbol{Z}, \boldsymbol{\gamma}^*)^2 \right] = \tilde{\mathcal{O}}(1)$ and it leads to our desired conclusion. □

Next, we obtain the following result on test error.

**Lemma D.2.** *With probability at least* $0.995$, *it holds that* $\mathcal{R}_{N_{\text{pt}}}(\boldsymbol{\gamma}^*, \boldsymbol{u}^*, \boldsymbol{v}^*, \boldsymbol{a}^*) - \tau = o(1)$.

*Proof of Lemma D.2.* From Lemma C.3, with probability at least $0.999$, we can choose $U = \tilde{\mathcal{O}}(r^{3\text{ge}(g_*)/2} m^{-1/2})$ such that $\boldsymbol{u}^* \leq U$ and we have

$$\mathcal{R}_{N_{\text{pt}}}(\boldsymbol{\gamma}^*, \boldsymbol{u}^*, \boldsymbol{v}^*, \boldsymbol{a}^*) - \tau$$

$$= \frac{1}{T_2} \sum_{t=T_1+1}^{T_1+T_2} \left| y^t - f\left(\boldsymbol{Z}^t, \boldsymbol{\gamma}^*, \boldsymbol{u}^*, \boldsymbol{v}^*, \boldsymbol{a}^*\right) \right|$$

$$+ \left( \mathcal{R}_{N_{\text{pt}}}(\boldsymbol{\gamma}^*, \boldsymbol{u}^*, \boldsymbol{v}^*, \boldsymbol{a}^*) - \frac{1}{T_2} \sum_{t=T_1+1}^{T_1+T_2} \left| y^t - f\left(\boldsymbol{Z}^t, \boldsymbol{\gamma}^*, \boldsymbol{u}^*, \boldsymbol{v}^*, \boldsymbol{a}^*\right) \right| \right) - \tau$$

$$\leq \sup_{\tilde{f}\in\tilde{\mathcal{F}}_U} \left( \mathcal{R}_{N_{\mathrm{pt}}}(\boldsymbol{\gamma}^*, \boldsymbol{u}^*, \boldsymbol{v}^*, \boldsymbol{a}^*) - \frac{1}{T_2}\sum_{t=T_1+1}^{T_1+T_2} \left| y^t - f\left(\boldsymbol{Z}^t, \boldsymbol{\gamma}^*, \boldsymbol{u}^*, \boldsymbol{v}^*, \boldsymbol{a}^*\right) \right| \right) + o(1),$$

where $\tilde{\mathcal{F}}_U := \mathcal{F}_U \cup \{(\boldsymbol{Z}, y) \mapsto y\}$. Using the standard symmetrization argument (Proposition 4.2 in Bach (2024)), we have

$$\mathbb{E}\left[ \sup_{\tilde{f}\in\mathcal{F}_U} \left( \mathcal{R}_{N_{\mathrm{pt}}}(\boldsymbol{\gamma}^*, \boldsymbol{u}^*, \boldsymbol{v}^*, \boldsymbol{a}^*) - \frac{1}{T_2}\sum_{t=T_1+1}^{T_1+T_2} \left| y^t - f\left(\boldsymbol{Z}^t, \boldsymbol{\gamma}^*, \boldsymbol{u}^*, \boldsymbol{v}^*, \boldsymbol{a}^*\right) \right| \right) \right]$$

$$\leq 2\mathrm{Rad}_{T_2}(\tilde{\mathcal{F}}_U)$$

$$\leq 2\mathrm{Rad}_{T_w}(\mathcal{F}_U) + \frac{2}{T_2}\mathbb{E}_{\substack{(\boldsymbol{Z}^t,y^t)\sim\mathcal{D}(N_{\mathrm{pt}}) \\ \boldsymbol{\epsilon}\sim\mathrm{Unif}(\{\pm 1\}^{T_2})}} \left[ \sum_{t=1}^{T_2} \left| \boldsymbol{\epsilon}[t]y^t \right| \right],$$

where the second inequality holds since $\mathcal{F}_U$ contains zero function. By the Cauchy-Schwarz inequality, we can also bound the second term as

$$\mathbb{E}_{\substack{(\boldsymbol{Z}^t,y^t)\sim\mathcal{D}(N_{\mathrm{pt}}) \\ \boldsymbol{\epsilon}\sim\mathrm{Unif}(\{\pm 1\}^{T_2})}} \left[ \sum_{t=1}^{T_2} \left| \boldsymbol{\epsilon}[t]y^t \right| \right] \leq \left( \mathbb{E}_{\substack{(\boldsymbol{Z}^t,y^t)\sim\mathcal{D}(N_{\mathrm{pt}}) \\ \boldsymbol{\epsilon}\sim\mathrm{Unif}(\{\pm 1\}^{T_2})}} \left[ \left( \sum_{t=1}^{T_2} \boldsymbol{\epsilon}[t]y^t \right)^2 \right] \right)^{\frac{1}{2}}$$

$$= \sqrt{T_2} \left( \mathbb{E}_{(\boldsymbol{Z},y)\sim\mathcal{D}(N_{\mathrm{pt}})} \left[ \left(y^t\right)^2 \right] \right)^{\frac{1}{2}}.$$

Combining with Lemma D.1, we have

$$\mathbb{E}\left[ \sup_{\tilde{f}\in\mathcal{F}_U} \left( \mathcal{R}_{N_{\mathrm{pt}}}(\boldsymbol{\gamma}^*, \boldsymbol{u}^*, \boldsymbol{v}^*, \boldsymbol{a}^*) - \frac{1}{T_2}\sum_{t=T_1+1}^{T_1+T_2} \left| y^t - f\left(\boldsymbol{Z}^t, \boldsymbol{\gamma}^*, \boldsymbol{u}^*, \boldsymbol{v}^*, \boldsymbol{a}^*\right) \right| \right) \right]$$

$$= \tilde{\mathcal{O}}\left( r^{3\mathrm{ge}(g_*)/2} T_2^{-1/2} \right) = o(1).$$

Note that $\sup_{\tilde{f}\in\tilde{\mathcal{F}}_U} \left( \mathcal{R}_{N_{\mathrm{pt}}}(\boldsymbol{\gamma}^*, \boldsymbol{u}^*, \boldsymbol{v}^*, \boldsymbol{a}^*) - \frac{1}{T_2}\sum_{t=T_1+1}^{T_1+T_2} \left| y^t - f\left(\boldsymbol{Z}^t, \boldsymbol{\gamma}^*, \boldsymbol{u}^*, \boldsymbol{v}^*, \boldsymbol{a}^*\right) \right| \right)$ is always non-negative due to $(\boldsymbol{Z}, y) \mapsto y \in \tilde{\mathcal{F}}_U$. Therefore, by applying Markov's inequality, we conclude that with probability at least 0.995,

$$\mathcal{R}_{N_{\mathrm{pt}}}(\boldsymbol{\gamma}^*, \boldsymbol{u}^*, \boldsymbol{v}^*, \boldsymbol{a}^*) - \tau = o(1).$$

$\square$

### D.2 TEST ERROR FOR PROMPTS WITH GENERAL LENGTH

For the last step, we extend the result of the test error to a general test time context length $N_{\mathrm{test}} = \tilde{\Omega}\left( r^{3\mathrm{ge}(g_*)} \right)$.

*Proof of Theorem 3.3.* To use the result of Lemma D.2, we bound the following quantity:

$$\left| \mathcal{R}_{N_{\mathrm{test}}}(\boldsymbol{\gamma}^*, \boldsymbol{u}^*, \boldsymbol{v}^*, \boldsymbol{a}^*) - \mathcal{R}_{N_{\mathrm{pt}}}(\boldsymbol{\gamma}^*, \boldsymbol{u}^*, \boldsymbol{v}^*, \boldsymbol{a}^*) \right|$$

$$= \left| \mathbb{E}_{(\boldsymbol{Z},y)\sim\mathcal{D}(N^*)} \left[ \left| y - f(\boldsymbol{Z}_{N_{\mathrm{pt}}}, \boldsymbol{\gamma}^*, \boldsymbol{u}^*, \boldsymbol{v}^*, \boldsymbol{a}^*) \right| - \left| y - f(\boldsymbol{Z}_{N_{\mathrm{test}}}, \boldsymbol{\gamma}^*, \boldsymbol{u}^*, \boldsymbol{v}^*, \boldsymbol{a}^*) \right| \right] \right|$$

$$\leq \mathbb{E}_{(\boldsymbol{Z},y)\sim\mathcal{D}(N^*)} \left[ \left| f(\boldsymbol{Z}_{N_{\mathrm{pt}}}, \boldsymbol{\gamma}^*, \boldsymbol{u}^*, \boldsymbol{v}^*, \boldsymbol{a}^*) - f(\boldsymbol{Z}_{N_{\mathrm{test}}}, \boldsymbol{\gamma}^*, \boldsymbol{u}^*, \boldsymbol{v}^*, \boldsymbol{a}^*) \right| \right].$$

Here, $Z_{N_{\mathrm{pt}}}$ and $Z_{N_{\mathrm{test}}}$ are input embeddings consisting of the first $N_{\mathrm{pt}}$ and $N_{\mathrm{test}}$ context examples, respectively, along with the same query $\boldsymbol{x}$, when given a prompt $\boldsymbol{Z}$. From Lemma A.1, the following holds with high probability:

$$\left| N_{\mathrm{pt}}^{-1}\mathsf{Mamba}(\boldsymbol{Z}_{\mathrm{pt}}; \boldsymbol{\gamma}^*) - N_{\mathrm{test}}^{-1}\mathsf{Mamba}(\boldsymbol{Z}_{\mathrm{test}}; \boldsymbol{\gamma}^*) \right| = o\left( r^{-3\mathrm{ge}(g_*)/2}(\log d)^{-2\deg(g_*)+2-C_{P_2}} \right).$$

Combining with Lipschitz continuity of ReLU, this implies

$$\left| f(\boldsymbol{Z}_{N_{\mathrm{pt}}}, \boldsymbol{\gamma}^*, \boldsymbol{u}^*, \boldsymbol{v}^*, \boldsymbol{a}^*) - f(\boldsymbol{Z}_{N_{\mathrm{test}}}, \boldsymbol{\gamma}^*, \boldsymbol{u}^*, \boldsymbol{v}^*, \boldsymbol{a}^*) \right|$$

$$\leq \sum_{j=1}^{m} |\boldsymbol{u}^*[j]| \left| N_{\mathrm{pt}}^{-1}\mathsf{Mamba}(\boldsymbol{Z}_{\mathrm{pt}};\boldsymbol{\gamma}^*) - N_{\mathrm{test}}^{-1}\mathsf{Mamba}(\boldsymbol{Z}_{\mathrm{test}};\boldsymbol{\gamma}^*) \right|$$

$$\leq \|\boldsymbol{u}\|\, m^{1/2} \left| N_{\mathrm{pt}}^{-1}\mathsf{Mamba}(\boldsymbol{Z}_{\mathrm{pt}};\boldsymbol{\gamma}^*) - N_{\mathrm{test}}^{-1}\mathsf{Mamba}(\boldsymbol{Z}_{\mathrm{test}};\boldsymbol{\gamma}^*) \right|$$

$$= \tilde{\mathcal{O}}(r^{3\mathrm{ge}(g_*)/2}m^{-1/2}) \cdot m^{1/2} \cdot o\left( r^{-3\mathrm{ge}(g_*)/2}(\log d)^{-2\deg(g_*)+2-C_{P_2}} \right)$$

$$= o(1),$$

where we apply the Cauchy-Schwarz inequality for the last inequality, and the last equality holds since we can make $C_{P_2}$ arbitrarily large. Therefore, we have

$$\left| \mathcal{R}_{N_{\mathrm{test}}}(\boldsymbol{\gamma}^*,\boldsymbol{u}^*,\boldsymbol{v}^*,\boldsymbol{a}^*) - \mathcal{R}_{N_{\mathrm{pt}}}(\boldsymbol{\gamma}^*,\boldsymbol{u}^*,\boldsymbol{v}^*,\boldsymbol{a}^*) \right| = o(1),$$

and this implies that our desired conclusion holds with probability at least 0.99. $\qquad\square$

