# OpenReview forum: "Mamba Can Learn Low-Dimensional Targets In-Context via Test-Time Feature Learning"
_ICLR.cc/2026/Conference — Submitted to ICLR 2026_

### Official Review · Reviewer_wSHm · 2025-10-27

**Soundness:** 3
**Presentation:** 3
**Contribution:** 2
**Rating:** 4
**Confidence:** 3

**Summary:**

The paper studies the in-context learning of Mamba model in a theoretical way. It provides theoretical sample complexity bound on the number of pretraining tasks to achieve a fixed ICL test error via a two-stage gradient pretraining scheme. The paper also proves Mamba can achieve test time feature learning and conducts experiment to verify this.

**Strengths:**

1. The paper theoretically studies the ICL of Mamba, which is an interesting topic.
2. The paper is the first to provide a sample complexity result on the ICL of Mamba.
3. The writing is clear which makes the paper easy to follow.

**Weaknesses:**

1. Since the authors listed proposition 4.1 as one of the core contributions, I think a formal version of it could benefit the paper, because currently it's hard to understand what $P_1$ and $P_2$ stands for without referring to the appendix (Lemma C.1).
2. The experiment doesn't collaborate with the theoretical results well. It only shows that Mamba **can** achieve in-context feature learning, which is not that surprising given empirical results on the ICL of Mamba models [1,2]. Perhaps some trends toward how the number of pretraining tasks affects the final ICL test error could be more relative to the point.
3. I'm currently not so sure about the siginificance of contribution compared with [3]. The two stage learning algorithm and the learning objective in this paper is the same as in [3], and the replacement of a transformer with a Mamba doesn't seem very fundamental.


[1] Grazzi, R., Siems, J., Schrodi, S., Brox, T., & Hutter, F. (2024). Is mamba capable of in-context learning?. arXiv preprint arXiv:2402.03170.

[2] Park, J., Park, J., Xiong, Z., Lee, N., Cho, J., Oymak, S., ... & Papailiopoulos, D. (2024). Can mamba learn how to learn? a comparative study on in-context learning tasks. arXiv preprint arXiv:2402.04248.

[3] Nishikawa, N., Song, Y., Oko, K., Wu, D., & Suzuki, T. Nonlinear transformers can perform inference-time feature learning. In Forty-second International Conference on Machine Learning.

**Questions:**

1. Could the authors ellaborate on the fundamental difference between your paper and [1]? Though I haven't read [1] carefully, but it seems to me that your work inherits the target function $f_*$, the learning objective loss $L$, the two-stage pretraining algorithm and the diagonal structure of $W_B^\top W_C$ (which in [1]'s case is $W_Q^\top W_K$, and the $1$ in the right bottom corner could be ommitted because Mamba doesn't have value matrix $W_V$). The main difference is the replacement of the transformer model with the Mamba model, which mimics a linear self attention (admittedly the nonlinear gating may add some difficulty). Overall, I'm concerned about the techinical contribution of this paper and would be willing to reconsider my rating if the authors could convince me of it.


[1] Nishikawa, N., Song, Y., Oko, K., Wu, D., & Suzuki, T. Nonlinear transformers can perform inference-time feature learning. In Forty-second International Conference on Machine Learning.

---

> ### Author Response · Authors · 2025-11-21
>
> We express our gratitude for your valuable comments. In the following, we address the points raised by the reviewer.
>
> ## **W1. Suggestion Regarding Proposition 4.1**
>
> We thank the reviewer for their valuable suggestion regarding Proposition 4.1. Following this suggestion, we have added a reference to its formal version, Proposition B.5. We have also revised our draft to explicitly clarify that $P_1$ and $P_2$ are independent of data.
>
> ## **W2. Experiments**
>
> We would like to first clarify that our main focus is theoretical studies, whereas [1] and [2] focus on extensive experiments on Mamba’s ICL capabilities using simple function classes. In addition, single-index models are not addressed in those works. Therefore, our experimental results are not trivially expectable from previous works, and both our theory and experimental results are valuable for understanding Mamba’s capabilities. However, we acknowledge that further experiments can be helpful for corroborate our theoretical findings. We have added new experiments to our revision (Section 6, Figure 1) varying not only the ambient dimension $d$, but also the intrinsic dimension $r$ and the pretraining context length $N_{\mathrm{pt}}$.
> These results align with our two key findings: (1) Mamba’s test-time sample complexity is comparable to that of Transformers and primarily depends on the intrinsic dimension $r$ rather than the ambient dimension $d$, and (2) Mamba has a smaller pretraining sample complexity than Transformers. We believe that these results corroborate our theoretical findings and that this addition significantly improves our paper.
>
> ## **W3, Q1 Comparison to Nishikawa et al. (2025)**
>
> We acknowledge that our work and [3] share some problem settings, such as the (Transformer, Mamba)-MLP structure and layerwise training. The second stage of our pretraining and [3] are similar, as in both cases, the MLP layer's role is to fit the link function $g_*$ given that the pretrained inner layer performs test-time feature learning. However, for the first stage of pretraining, our work has significant technical distinctions from [3]. Crucially, the source of nonlinearity in Mamba is fundamentally different from the nonlinearity in Softmax Transformers, making the existing analysis in [3] inapplicable to our setting.
>
> First of all, since the nonlinearities introduced by Mamba and the Transformer are different (gating and Softmax, respectively), it is nontrivial to prove that the gating mechanism can also reduce the information exponent to the generative exponent.
> In addition, the MLP layer plays a fundamentally different role in our work than in [3] during the analysis of Stage 1 (as we described in the “Improved Sample Complexity of Pretraining” paragraph in Section 4.1). In our analysis, the MLP's nonlinearity is crucial: without it, the expectation $\mathbb{E} [ y C_\beta \odot \phi(x)]$ (which appears in the updated parameter $\gamma^* $) would be zero for cases where $\mathrm{ie}(g_*)>2$. This is because the input embedding $\phi(x)$ only contains up to second-order polynomials of the input $x$. In contrast, the analysis of [3] also works when considering a regime where nonlinearity does not have a role (i.e., the term inside ReLU is positive), which is possible in their case because Softmax generates the necessary higher-order functions of $x$. Due to this significant difference, we carefully address the nonlinearity of the MLP using totally independent arguments (Lemma B.3, Lemma B.4). As a result, our analysis achieves an improved pretraining sample complexity compared to [3].
>
> We hope this detailed comparison and clarification sufficiently address the reviewer's concerns regarding the novelty of our technical contributions.
>
> We hope our response helps to resolve any concerns.
>
> Best regards,
>
> Authors
>
> ---
>
> Reference
>
> [1] Grazzi et al. Is Mamba Capable of In-Context Learning? In AutoML, 2024.
>
> [2] Park et al. Can Mamba Learn How to Learn?: a Comparative Study on In-Context Learning Tasks. In ICML, 2024.
>
> [3] Nishikawa et al. Nonlinear transformers can perform inference-time feature learning. In ICML, 2025.

---

### Official Review · Reviewer_cryr · 2025-10-29

**Soundness:** 3
**Presentation:** 2
**Contribution:** 3
**Rating:** 6
**Confidence:** 4

**Summary:**

This paper studies the ICL ability of Mamba on single-index models by characterizing the optimization dynamics and the sample complexity in pretraining and testing. The theoretical results indicate that Mamba can implement test-time feature learning, which is enabled by the nonlinear gating. The authors also make a comparison between their results and previous works and other archtectures.

**Strengths:**

1. The theoretical analysis is quite strong and solid.

2. The studied problem on Mamba with single-index models is interesting and novel.

**Weaknesses:**

1. The illustration in Section 4 is confusing to me.

(a) Overall, I think many of this section is not a "proof overview" but some important discussion that you can put in previous sections. In other words, in Section 4.1, you introduce too much about some theoretical conclusions or the differences from existing works rather than the logic chain to derive your main theorem, which should be the proof idea from my understanding.

(b). Regarding the role of gating and input embedding, $c_\beta$ vanishes when $ie(g_*)>2$ if it is linear attention, but what about Mamba? Do you mean $c_\beta$ will not vanish in this case? When I first read it, I got confused because I thought you would directly answer this question, but you started to talk about the generative exponent. I think it is better to first answer this question or briefly summarize the role of gating at the beginning of this paragraph.

(c). Proposition 4.1 is still not "informal" enough. Why not directly write $Mamba(\cdot)$ as a $g_*(\cdot)$ function in Eqn. 2? You mention this possibility in line 433 that $g_*(z)$ is a polynomial of $z^{ge(g_*)}$. By the way, I took some time to recall what $P_1$ and $P_2$ are in this proposition. I found they are defined in line 300. It is better to intuitively explain them around this proposition.

2. Not enough experiments are presented. Can you empirically show the advantages of Mamba in sample complexity compared with Transformers, and can you empirically relate it to the nonlinear gating part?

**Questions:**

1. Why do you need an MLP layer in your analysis? In line 365, you say the improvement of sample complexity comes from the MLP, then you need to clarify what makes Mamba better than Transformers in some aspect because MLP is not a key difference between Mamba and Transformers from my understanding. You need to distinguish what improvement comes from MLP and what comes from the key component of Mamba, i.e, the nonlinear gating.

2. Why do you formulate $W_B^\top W_C$ with a simple veector $\gamma$ and why do you initialize $\gamma$ in a sepcial way in line 256?

3. It seems you reduce the analysis of a $d+1$-dimensional $w$ to a scalar $\rho$. What is the reason?

4. Why do you formulate $\phi(\theta)$ in a special way in line 154? Can you handle of case of $\phi(x)=x$?

---

> ### Author Response · Authors · 2025-11-21
>
> We would like to express our appreciation for your valuable comments. In the following, we address the points raised by the reviewer.
>
> ## **W1 Revision of Section 4.1**
>
> We thank the reviewer for the constructive suggestion to improve the Proof Overview section, particularly Section 4.1. Following this suggestion, we have revised this section. We now begin with the outcome of Mamba layer training (Proposition 4.1) and then move on to explain the crucial role of nonlinearity in the gating and MLP, a part we have also improved based on the reviewer's feedback.
>
> We would also like to clarify the concerns regarding Proposition 4.1. We apologize for the confusion caused by this "abuse of notation." In our revision, we now explicitly clarify that $P_1$ and $P_2$ are quantities independent of data and are distinct from the pushforward measures defined in Section 3.1. To correct this, we have updated the notation for the pushforward measures to $\mathbb{P}$.
>
> We would further clarify the meaning of Proposition 4.1. It implies that the pretrained Mamba's role is to extract features. More precisely, it outputs the projection of the query onto the feature vector, raised to the power of $\mathrm{ge}(g_*)$. Proposition 4.1 does not state $\mathrm{Mamba}(\cdot)=g_*(\cdot)$ directly because our analysis isolates only the leading-order dependency on the projection $\langle x,\beta\rangle$, rather than the entire link function. At this stage, the model does not learn information about the link function $g_*$; that task is handled by the MLP layer in the second stage of training. The proof for the MLP part then relies on the fact that $g_*(z)$ is a polynomial of $z^{\mathrm{ge}(g_*)}$.
>
> ## **Q1 Clarification of the Role of MLP**
>
> In our analysis, the MLP layer serves two crucial roles. The first, as we addressed above, is fitting the link function $g_*$ during the second stage of training. The second role comes into play during the first stage of training and we discuss this in “Improved Sample Complexity of Pretraining” paragraph in Section 4.1.
>
> Without the nonlinearity in the MLP, the expectation $\mathbb{E} [ y C_\beta \odot \phi(x)]$ (which appears in the updated parameter $ \gamma^* $ ) would be zero for cases where $\mathrm{ie}(g_*)>2$. This is because the input embedding $\phi(x)$ only contains up to second-order polynomials of the input $x$. The analysis of [1] also works when considering a regime where nonlinearity does not have a role (i.e., the term inside ReLU is positive), which is possible in their case because Softmax generates the necessary higher-order functions of $x$.
> We have added this discussion and clarified the distinct roles of the two nonlinearities in gating and MLP in our revision (both Section 4.1 and 4.2).
>
> ## **W2 Further Experiments**
>
> Thank you for your suggestion.
>
> We have added new experiments to our revision (Section 6, Figure 1) varying not only the ambient dimension $d$, but also the intrinsic dimension $r$ and the pretraining context length $N_{\mathrm{pt}}$.
> These results align with our two key findings: (1) Mamba’s test-time sample complexity is comparable to that of Transformers and primarily depends on the intrinsic dimension $r$ rather than the ambient dimension $d$, and (2) Mamba has a smaller pretraining sample complexity than Transformers. We believe that these results corroborate our theoretical findings and that this addition significantly improves our paper.
>
> As we mentioned above, our theoretical analysis identifies the MLP's nonlinearity (not the gating mechanism) as the key factor enabling Mamba to achieve better pretraining sample complexity than the Transformer within the theoretical framework adopted in our work and [1]. However, this raises an interesting question, as the practical Transformers used in our experiments still required more samples. The reason for this difference is currently unclear. We believe that investigating whether Transformers can also achieve a pretraining sample complexity independent of $\mathrm{ie}(g_*)$ would be an interesting research direction for understanding the limitations of Transformer.

---

> ### Author Response · Authors · 2025-11-21
>
> ## **Q2 Simplification of Mamba Parameter**
>
> We merge $W_B^\top W_C$ into a single matrix and sparsify it for tractable optimization dynamics. This is a standard approach widely adopted in the theory of ICL literature [2]-[5]. We further simplify this matrix to be diagonal (parameterized by a vector $\gamma$), as this structure is sufficient to efficiently learn our target function.
>
> In addition, we use the specific initialization scheme for $\gamma$ described in Stage I of Algorithm 1. This specific choice is made to theoretically control the nonlinearity of the MLP layer (as discussed above). We would like to emphasize that this is one possible choice to facilitate our analysis, and other choices are also possible.
>
> Our assumptions can be relaxed in some sense, such as sparsifying $W_B$ and $W_C$ into diagonal matrices without merging and using a more general initialization. However, this would add complexity without changing the main theoretical insights.
>
> ## **Q3. Simplification of gating part**
>
> In our analysis, the gating mechanism plays a crucial role because it performs a nonlinear transformation on the context labels. As label transformation is the essential component, our analysis also holds for cases where the $d$-dimensional components of $w$ that correspond to the inputs are non-zero, as can be shown with a similar argument. However, for clarity and simplicity, we focused on the simplest case, which we believe sufficiently captures the essential mechanism of this analysis.
>
> ## **Q4. Input embedding**
>
> As we discussed in the “The Role of Gating and Input Embedding” paragraph in Section 4, our result can also be shown to hold with $\phi(x) = x$ (which induces the standard input embedding) for the case of $\mathrm{ge}(g_*) = 1$. Our specific design for the input embedding is chosen to address the $\mathrm{ge}(g_*)=2$ case, which requires a 2nd-order function of $x$. We believe that any other embedding containing 1st and 2nd-order functions of the input could also work, but we chose the simplest one.
>
> We hope our response helps to resolve any concerns and confusion.
>
> Best regards,
>
> Authors
>
> ---
>
> Reference
>
> [1] Nishikawa et al. Nonlinear transformers can perform inference-time feature learning. In ICML, 2025.
>
> [2] Ahn et al. Transformers learn to implement preconditioned gradient descent for in-context learning. In NeurIPS, 2023.
>
> [3] Zhang et al. Trained transformers learn linear models in-context. JMLR, 2024.
>
> [4] Mahankali et al. One step of gradient descent is provably the optimal in-context learner with one lyaer of linear self-attention. ICLR, 2024.
>
> [5] Kim & Suzuki. Transformers learn nonlinear features in context: Nonconvex mean-field dynamics on the attention landscape. ICML, 2024.

---

### Official Review · Reviewer_XMDG · 2025-11-01

**Soundness:** 3
**Presentation:** 3
**Contribution:** 2
**Rating:** 4
**Confidence:** 4

**Summary:**

This paper provides a theoretical analysis of the Mamba architecture’s in-context learning (ICL) capability, focusing on low-dimensional nonlinear target functions. The authors establish that Mamba, when pretrained via a gradient-based algorithm, can achieve efficient in-context feature learning at test time. They prove that Mamba extracts task-relevant features directly from contextual examples, yielding sample complexity comparable to nonlinear Transformers while improving upon linear kernel-based models.

**Strengths:**

1. Most existing theoretical works on understanding in-context learning (ICL) focus on Transformers, and this paper’s analysis of Mamba—along with its comparison to Transformers—offers valuable and insightful contributions.

2. Provide conceptual insight: The identification of Mamba’s nonlinear gating as the driver of test-time feature learning is conceptually meaningful for understanding gated recurrent architectures.

**Weaknesses:**

1. Algorithm 1 appears unnatural. Algorithm 1 optimize the Mamba layer and the MLP layer separately, which is not how neural networks are actually optimized in practice. This mismatch with realistic training procedures weakens the contribution of the paper.

2. The input embedding also appears unnatural and overly hand-crafted, raising doubts about whether real models could actually learn such embeddings in practice.

**Questions:**

1. Could the authors consider analyzing Mamba’s training convergence under a more practical optimization process? (Weakness 1)

2. Could the authors provide an analysis of generalization—for example, how the model behaves when the data distribution at test time differs from that during training?

---

> ### Author Response · Authors · 2025-11-21
>
> Thank you for your valuable comments. In the following, we address the points raised by the reviewer.
>
> ## **W1, Q1 Pretraining Algorithm**
>
> Our simplified layer-wise algorithm—which employs one-step gradient descent for the inner layer and ridge regression for the outer layer—is specifically designed to make the analysis tractable. This approach is standard and has been widely adopted in previous theoretical works including [1]-[4], and we emphasize that the analysis remains highly non-trivial even with this simplification.
>
> Importantly, all our experiments train Mamba end-to-end using standard optimization, not Algorithm 1. Algorithm 1 is introduced only for theoretical tractability, following prior feature-learning papers. Crucially, the theoretical behaviors predicted under Algorithm 1—such as test-time feature extraction, scaling in intrinsic dimension $r$, and improved pretraining sample complexity—are all empirically observed under realistic end-to-end training. We believe that analysis within this simplification still provides valuable theoretical insights for understanding Mamba’s ICL mechanism and an extension to more practical optimization methods would require more advanced techniques.
>
> ## **W2 Choice of Input Embedding**
>
> As we discussed in the “The Role of Gating and Input Embedding” paragraph in Section 4, our result can also be shown to hold with $\phi(x) = x$ (which induces the standard input embedding) for the case of $\mathrm{ge}(g_*) = 1$. Our specific design for the input embedding is chosen to address the $\mathrm{ge}(g_*)=2$ case, which requires a high-order function of $x$ as input. We would like to emphasize that this embedding can be implemented with a practical mechanism such as a simple version of a Gated Linear Unit (GLU) [5]. In addition, we believe that any other embedding containing 1st and 2nd-order functions of the input could also work, and that these necessary features could be extracted by nonlinear layers in a more practical scenario.
>
> ## **Q2 OOD Generalization**
>
> We would like to clarify that the data distribution within a test prompt is inherently different from that of the pretraining tasks, as the feature vectors for each task are randomly sampled.
>
> We can further consider an even more general case, such as a different link function ($g_*$) between pretraining and test-time. In this scenario, we believe that generalization can still be achieved with test-time training. Our analysis revealed that the pretrained Mamba can extract feature vectors in-context, and the primary role of the MLP is to fit the link function. Therefore, the MLP could be efficiently fitted to this new link function at test-time using the provided context examples, while we leave rigorous analysis as future works.
>
> We hope our response helps to resolve any concerns and confusion.
>
> Best regards,
>
> Authors
>
> ---
>
> Reference
>
> [1] Ba et al. High-dimensional asymptotics of feature learning: How one gradient step improves the representation. In NeurIPS, 2022.
>
> [2] Damian et al. Neural networks can learn representations with gradient descent. In COLT, 2022.
>
> [3] Oko et al. Pretrained Transformer Efficiently Learns Low-Dimensional Target Functions In-Context. In NeurIPS, 2024.
>
> [4] Nishikawa et al. Nonlinear transformer perform inference-time feature learning. In ICML, 2025.
>
> [5] Sun et al. On the role of transformer feed-forward layers in nonlinear in-context learning. arXiv, 2025.

---

### Official Review · Reviewer_5eCn · 2025-11-01

**Soundness:** 3
**Presentation:** 3
**Contribution:** 3
**Rating:** 6
**Confidence:** 2

**Summary:**

This paper analyzes Mamba's in-context learning on Gaussian single-index models and proves that pretrained Mamba achieves test-time feature learning by gating operation. This enables test-time sample complexity comparable to nonlinear Transformers, and better than linear-Transformers. The main theorem specifies pretrain and context lengths, number of tasks and width needed to achieve low test error with high probability. Experiments with r=8, d=16,32 show comparable performance of Transformer and Mamba, and that the results are insensitive to d. Both models outperform kernel ridge regression even when the kernel operates on the intrinsic features, supporting the main theory of test-time feature learning.

**Strengths:**

- The paper shows that the nonlinear gate changes the label statistic so the task becomes identifiable. This allows the dependence of the sample complexity on the true nonlinearity of the teacher function.
- The main theorem gives concrete conditions under which a one-layer Mamba with MLP head can learn features from the prompt at test time, and the required number of in-context examples scales with the intrinsic dimension of the task subspace.
- The detailed analysis and comparison with not only the Transformer architecture but also kernel methods, explain the benefits of gating.

**Weaknesses:**

- Although not a major weakness, the distributional assumptions are strong: Gaussian single-index models with polynomial links and Hermite features.
- The experiments can be conducted over more range of intrinsic dimensions or sample sizes.

**Questions:**

- Which steps in the argument strictly require the Gaussian inputs and a polynomial link? Can this work be extended to non-Gaussian, or non-polynomial cases, with heavy-tailed noise?
- How do the required prompt lengths and task counts scale as the intrinsic dimension increases?

---

> ### Author Response · Authors · 2025-11-21
>
> We would like to express our gratitude for your valuable comments. In the following, we address the points raised by the reviewer.
>
> ## **W1, Q1 Regarding our problem setting**
>
> While our assumptions follow the standard setting used across recent single-index model literature, we acknowledge the reviewer’s concern and clarify below why they are reasonable and how they can be relaxed. However, we would like to emphasize that the theoretical analysis even under this setting remains highly non-trivial. We also wish to clarify that the Hermite features arise automatically from expanding the link function under the Gaussian measure; they are not an assumption about the model or architecture.
>
> Indeed, many previous works including [1]-[5] have significantly advanced the theoretical analysis of neural network learning within this framework. Our paper builds upon this line of research by extending the analysis to the ICL capabilities of Mamba.
>
> The Gaussian input assumption allows us to utilize established technical results, including the information exponent and generative exponent, which stem from previous literature focusing on Gaussian data. These results can be established because Gaussian input is known to induce a benign loss landscape.
> Importantly, as [6] shows, this 'benign landscape' property continues to hold for broader data distributions, including some spherically symmetric and quasi-symmetric distributions.
>
> Therefore, we believe that the Gaussian assumption for the theoretical tools we use can be relaxed for more general cases without changing the essential message. However, rigorously proving this generalization would require a separate line of research.
>
> Given this, and since our primary target is understanding Mamba’s ICL capability, we chose the Gaussian setting because it enables a clean, interpretable analysis of Mamba’s gating mechanism; extending the proof to general distributions would require additional machinery but not change the core insight.
>
> An extension of our result to non-polynomial link functions or heavy-tailed noise is non-trivial, as our analysis relies on technical results (like Lemma 3.2) that are specifically guaranteed for polynomial link functions, and on the argument that population loss can be approximated by finite samples, which is not directly guaranteed for the heavy-tailed noise case. However, we believe that the main message from our analysis does not change, while a rigorous extension to these more general cases requires advanced techniques.
>
> ## **W2 More experimental results**
>
> Thank you for your suggestion. We have added new experiments to our revision (Section 6, Figure 1) varying not only the ambient dimension $d$, but also the intrinsic dimension $r$ and the pretraining context length $N_{\mathrm{pt}}$.
> These results align with our two key findings: (1) Mamba’s test-time sample complexity is comparable to that of Transformers and primarily depends on the intrinsic dimension $r$ rather than the ambient dimension $d$, and (2) Mamba has a smaller pretraining sample complexity than Transformers. We believe that these results corroborate our theoretical findings and that this addition significantly improves our paper.
>
> ## **Q2 Prompt length and task counts**
>
> We clarify that our main result (Theorem 3.3) provides explicit scales for pretraining that depend on both the intrinsic dimension $r$ and the ambient dimension $d$. As stated, these scales increase with $r$, and their exponents depend only on the generative exponent of $g_∗$. For clarity in our summary table (Table 1), we reported only the dominant terms (those involving $d$). The $r$ terms were omitted based on the fact that $r<d$.
>
> Thanks for your time and consideration.
>
> Best regards,
>
> Authors
>
> ---
> Reference
>
> [1] Ben Arous et al. Online stochastic gradient descent on non-convex losses from high-dimensional inference. JMLR, 2021.
>
> [2] Damian et al. Smoothing the Landscape Boosts the Signal for SGD: Optimal Sample Complexity for Learning Single Index Models. In NeurIPS, 2023.
>
> [3] Damian et al. Computational-Statistical Gaps in Gaussian Single-Index Models. In COLT, 2024.
>
> [4] Lee et al. Neural network learns low-dimensional polynomials with SGD near the information-theoretic limit. In NeursIPS, 2024.
>
> [5] Nishikawa et al. Nonlinear transformers can perform inference-time feature learning. In ICML, 2025
>
> [6] Bruna et al. On Single Index Models beyond Gaussian Data. In NeurIPS, 2023

---

### Author Response · Authors · 2025-11-21
**Summary of Revision**

We thank the reviewers for their time and valuable comments. We have updated our manuscript to reflect your feedback. The major revisions are summarized below and are marked in blue in the updated draft:
- We have added experimental results varying the intrinsic dimension $r$ and the pretraining context length $N_{\mathrm{pt}}$. These results align with our theoretical findings.
- We have restructured Section 4 to enhance the logical flow and readability. We now present the goal first, followed by a discussion on the role of each component.
- We have refined some parts of the paper to clarify points that were originally noted as vague.

We hope our revision helps to resolve any concerns and confusion.

Best regards,

Authors

---

### Meta-Review · Area_Chair_4ANE · 2026-01-14

**Summary:**

The paper provides a theoretical analysis for ICL capabilities of Mamba for single-index models. The theoretical results rely on distributional assumptions, such as Gaussian marginals, and also use a training algorithm that is different from those used in practice. The reviews raised questions about the writing (particularly the section on proof overview) and asked for more experiments. There were also some questions regarding the novelty of the work.

**Reviewer Concerns:**

The reviews raised questions about presentation, the strength of assumptions and emphasized the need for more experiments. Some of these have been addressed by the authors through rebuttals and revisions.

**Reviewer Scores:**

Since there is no reviewer interaction, I find it hard to comment on how the reviewers would have changed their scores.

---

### Decision · Program_Chairs · 2026-01-26

Reject